# Spectral Bias Outside the Training Set for Deep Networks in the Kernel Regime

**Benjamin Bowman**
UCLA Department of Mathematics
`benbowman314@math.ucla.edu`

**Guido Montúfar**
UCLA Departments of Mathematics and Statistics and MPI MIS
`montufar@math.ucla.edu`

## Abstract

We provide quantitative bounds measuring the $L^2$ difference in function space between the trajectory of a finite-width network trained on finitely many samples from the idealized kernel dynamics of infinite width and infinite data. An implication of the bounds is that the network is biased to learn the top eigenfunctions of the Neural Tangent Kernel not just on the training set but over the entire input space. This bias depends on the model architecture and input distribution alone and thus does not depend on the target function which does not need to be in the RKHS of the kernel. The result is valid for deep architectures with fully connected, convolutional, and residual layers. Furthermore the width does not need to grow polynomially with the number of samples in order to obtain high probability bounds up to a stopping time. The proof exploits the low-effective-rank property of the Fisher Information Matrix at initialization, which implies a low effective dimension of the model (far smaller than the number of parameters). We conclude that local capacity control from the low effective rank of the Fisher Information Matrix is still underexplored theoretically.

## 1 Introduction

Training heavily overparameterized networks via gradient based optimization has become standard operating procedure in deep learning. Overparameterized networks are able to interpolate arbitrary labels both in principle and in practice (Zhang et al., 2017), rendering classical PAC learning theory insufficient to explain the generalization of networks within this modality. The high capacity of modern networks ensures that there are both good and bad empirical risk minimizers. Miraculously the network preferentially chooses the good solutions and sidesteps those that are unfavorable, posing a challenge and opportunity to today's researchers.

The success of overparameterized networks has prompted the theoretical community to search for more subtle forms of capacity control (Neyshabur et al., 2015, 2017; Gunasekar et al., 2017). The contemporary point-of-view is that the data distribution, model parameterization, and optimization algorithm are all relevant in limiting complexity. This has led to a variety of efforts to characterize the properties that networks and related models are biased towards when optimized via gradient descent. Examples include max-margin bias for classification problems (Soudry et al., 2018; Ji & Telgarsky, 2019; Nacson et al., 2019; Gunasekar et al., 2018), minimum nuclear norm bias for matrix factorization (Gunasekar et al., 2017; Li et al., 2018; Gunasekar et al., 2018), and minimum RKHS norm bias in the kernel regime (Zhang et al., 2020).

36th Conference on Neural Information Processing Systems (NeurIPS 2022).

Empirically it is known that neural networks tend to learn low Fourier frequencies first and add higher frequencies only later in training (Rahaman et al., 2019; Xu et al., 2019; Yang et al., 2022), the phenomenon that has been titled "Spectral Bias" or the "Frequency Principle". Theoretical justifications of this have been proposed by studying networks in the kernel regime. For shallow univariate ReLU networks Basri et al. (2019, 2020) demonstrate that the dominant eigenfunctions of the Neural Tangent Kernel (NTK) (Jacot et al., 2018) correspond to the low Fourier frequencies for the uniform distribution and more generally to smoother components for nonuniform distributions. This echos the results by Williams et al. (2019) and Jin & Montúfar (2021) that show that univariate ReLU networks in the kernel regime are biased towards smooth interpolants. Abstracting away from Fourier frequencies, "Spectral Bias" can be interpreted more broadly to mean bias towards learning the top eigenfunctions of the Neural Tangent Kernel. By looking at empirical approximations to the eigenfunctions, spectral bias was demonstrated to hold on the training set by Arora et al. (2019a), Basri et al. (2020), and Cao et al. (2021). A recent work by Bowman & Montúfar (2022) was able to demonstrate that spectral bias holds off the training set for shallow feedforward networks when the network is underparameterized. In the present work we exploit the low-effective-rank property of the Fisher Information Matrix and are able to demonstrate that spectral bias holds outside the training set without the underparameterization requirement. In fact the number of samples can be on the same order as the width of the network. Furthermore, by leveraging a recent work by Liu et al. (2020b) bounding the Hessian of wide networks, our result permits deep networks with fully connected, convolutional, and residual layers. Consequently we are able to conclude that spectral bias holds for more realistic sample complexities and diverse architectures.

## 1.1 Our Contributions

- We provide quantitative bounds measuring the $L^2$ difference in function space between the trajectory of a finite-width network trained on finitely many samples from the idealized kernel dynamics of infinite width and infinite data (see Theorem 3.5 and Corollary 3.7).

- As an implication of these bounds, eigenfunctions of the NTK integral operator (not just their empirical approximations) are learned at rates corresponding to their eigenvalues (see Corollary 3.7 and Observation 3.8).

- We demonstrate that the network will inherit the bias of the kernel at the beginning of training even when the width only grows linearly with the number of samples (see Observation 3.9).

## 1.2 Related Work

**NTK Convergence Results** The NTK was introduced by Jacot et al. (2018) while almost concurrently Du et al. (2019b) used it implicitly to prove a global convergence guarantee for gradient descent applied to a shallow ReLU network. These two highly charismatic works led to a flurry of subsequent works, of which we can only hope to provide a partial list. Global convergence for arbitrary labels was addressed in a series of works (Du et al., 2019b,a; Oymak & Soltanolkotabi, 2020; Allen-Zhu et al., 2019; Nguyen & Mondelli, 2020; Nguyen, 2021; Zou et al., 2020; Zou & Gu, 2019). For arbitrary labels to our knowledge all works require the network width to either grow polynomially with the number of samples $n$ or the inverse desired accuracy $\epsilon^{-1}$. If one assumes the target function aligns with the NTK model, for shallow networks this can be reduced to polylogarithmic width for the logistic loss (Ji & Telgarsky, 2020) or linear width for the squared loss (E et al., 2020; Su & Yang, 2019; Bowman & Montúfar, 2022).

**Spectrum of the NTK/Hessian and Generalization** The fact that the NTK tends to have a small number of large outlier eigenvalues has been observed in many works (e.g. Arora et al. 2019a; Oymak et al. 2020; Li et al. 2020). Papyan (2020) demonstrated that for classification problems the logit gradients cluster within classes, which produces outliers in the spectra of the NTK and the Hessian of the loss. There have been a series of works analyzing the NTK/Hessian spectrum theoretically using random matrix theory and other tools (e.g. Karakida et al. 2021; Pennington & Worah 2018; Pennington & Bahri 2017; Fan & Wang 2020; Yang & Salman 2019). Recently the spectrum of the NTK integral operator for ReLU networks has been shown to asymptotically follow a power law (Velikanov & Yarotsky, 2021). Arora et al. (2019a) provided a generalization bound that is effective when the labels align with the top eigenvectors of the NTK. Oymak et al. (2020) were able to use the low effective rank of the NTK to obtain generalization bounds, and Li et al. (2020) used the same property to demonstrate robustness to label noise. The low effective rank of the Hessian has also been

incorporated into PAC-Bayes bounds, most recently by Yang et al. (2021). Interestingly, the notion of the effective dimension they define is essentially the same quantity we use to bound the model complexity of the network's linearization.

**NTK Eigenvector and Eigenfunction Convergence Rates**   Luo et al. (2020) explicitly tracked the dynamics of the infinite width shallow model in the Fourier domain. Arora et al. (2019a) demonstrated that when training the hidden layer of a shallow ReLU network, the residual error on the training set projected along eigenvectors of the NTK Gram matrix decays linearly at rates corresponding to the eigenvalues. Cao et al. (2021) proved a similar statement for training both layers, and Basri et al. (2020) proved the analogous statement for a deep fully connected ReLU network where the first and last layer are fixed. Our result can be viewed as the corresponding statement for the test residual instead of the empirical residual: projections of the test residual along eigen*functions* of the NTK *integral operator* are learned at rates corresponding to their eigenvalues. This was shown in a recent work (Bowman & Montúfar, 2022) for shallow fully connected networks that are underparameterized. By contrast our result does not require the network to be underparameterized, and holds for deep networks with fully connected, convolutional, and residual layers. We view our fundamental contribution as demonstrating that spectral bias holds with more realistic sample complexities and in considerable generality with respect to model architecture.

## 2   Preliminaries

### 2.1   Notation

Vectors $v \in \mathbb{R}^k$ will be column vectors by default. We will let $\langle \bullet, \bullet \rangle$ and $\|\bullet\|_2$ denote the Euclidean inner product and norm. We define $\langle \bullet, \bullet \rangle_{\mathbb{R}^n} = \frac{1}{n}\langle \bullet, \bullet \rangle$ and $\|\bullet\|_{\mathbb{R}^n} := \sqrt{\langle \bullet, \bullet \rangle_{\mathbb{R}^n}}$ to be the normalized Euclidean inner product and norm. The notation $\overline{B}(v, r) := \{w : \|w - v\|_2 \leq r\}$ will denote the *closed* Euclidean ball centered at $v$ of radius $r$. $\|A\|_{op} := \sup_{\|v\|_2 = 1} \|Av\|_2$ will denote the operator norm for matrices. For a symmetric matrix $A \in \mathbb{R}^{k \times k}$, $\lambda_i(A)$ denotes its $i$-th largest eigenvalue, i.e. $\lambda_1(A) \geq \lambda_2(A) \geq \cdots \geq \lambda_k(A)$. For a set $A$ we will let $|A|$ denote its cardinality. For a natural number $k \geq 1$, we will let $[k] := \{1, \ldots, k\}$. We will let $L^p(X, \nu)$ denote the $L^p$ space over domain $X$ with measure $\nu$. We will denote the inner product associated with $L^2(X, \nu)$ as $\langle \bullet, \bullet \rangle_\nu$. We will use the standard big $O$ and $\Omega$ notation with $\tilde{O}$ and $\tilde{\Omega}$ hiding logarithmic terms.

### 2.2   NTK Dynamics

Let $f(x; \theta)$ be our scalar-valued neural network model taking inputs $x \in X \subset \mathbb{R}^d$ parameterized by $\theta \in \mathbb{R}^p$. For now we will not specify a specific architecture. Our training data will be $n$ input-label pairs $\{(x_1, y_1), \ldots, (x_n, y_n)\} \subset \mathbb{R}^d \times \mathbb{R}$ where we assume that the labels $y_i$ are generated from a fixed scalar-valued target function $f^*$, namely $f^*(x_i) = y_i$. We will let $y \in \mathbb{R}^n$ denote the label vector $y = (y_1, \ldots, y_n)^T$. Let $\hat{r}(\theta) \in \mathbb{R}^n$ denote the vector that measures the residual error on the training set, whose $i$-th entry is $\hat{r}(\theta)_i := f(x_i; \theta) - y_i$. We will optimize the squared loss

$$\Phi(\theta) := \frac{1}{2n} \|\hat{r}(\theta)\|_2^2 = \frac{1}{2} \|\hat{r}(\theta)\|_{\mathbb{R}^n}^2$$

via gradient flow

$$\partial_t \theta_t = -\partial_\theta \Phi(\theta),$$

which is the continuous time analog of gradient descent. For conciseness we will denote $\hat{r}(\theta_t)$ by $\hat{r}_t$ and let $r_t(x) := f(x; \theta_t) - f^*(x)$ denote the residual for an arbitrary input $x$ not necessarily in the training set. We may also write $r(x; \theta) := f(x; \theta) - f^*(x)$ for the residual for an arbitrary $\theta$.

We recall some key definitions and facts about the NTK. For a comprehensive introduction we refer the reader to Jacot et al. (2018). We recall the definition of the analytical NTK

$$K^\infty(x, x') := \mathbb{E}_{\theta_0 \sim \mu}\left[\langle \nabla_\theta f(x; \theta_0), \nabla_\theta f(x'; \theta_0)\rangle\right],$$

where the expectation is taken over the parameter initialization $\theta_0 \sim \mu$. The kernel $K^\infty$ induces an integral operator $T_{K^\infty} : L^2(X, \rho) \to L^2(X, \rho)$

$$T_{K^\infty} g(x) := \int_X K^\infty(x, s)g(s)d\rho(s), \tag{1}$$

where $X$ is our input space and $\rho$ is the input distribution. We assume our training inputs $x_1, \ldots, x_n$ are i.i.d. samples from $\rho$. More generally, for a continuous kernel $K(x, x')$ we define $T_K : L^2(X, \rho) \to L^2(X, \rho)$

$$T_K g(x) := \int_X K(x, s)g(s)d\rho(s). \tag{2}$$

Returning back to $K^\infty$, by Mercer's theorem we have the decomposition

$$K^\infty(x, x') = \sum_{i=1}^\infty \sigma_i \phi_i(x)\phi_i(x'),$$

where $\{\phi_i\}$ is an orthonormal basis for $L^2(X, \rho)$ and $\{\sigma_i\}$ is a nonincreasing sequence of positive values. We will see that the bias at the beginning of training within our framework can be described entirely through the operator $T_{K^\infty}$ and its eigenfunctions. We note that $T_{K^\infty}$ depends only on the model architecture, parameter initialization distribution $\mu$, and input distribution $\rho$. The training data sample $x_1, \ldots, x_n$ introduces a discretization of the operator $T_{K^\infty}$

$$T_n g(x) := \frac{1}{n} \sum_{i=1}^n K^\infty(x, x_i)g(x_i) = \int_X K^\infty(x, s)g(s)d\widehat{\rho}(s), \tag{3}$$

where $\widehat{\rho} = \frac{1}{n} \sum_{i=1}^n \delta_{x_i}$ is the empirical measure. We now introduce the time-dependent NTK

$$K_t(x, x') := \langle \nabla_\theta f(x; \theta_t), \nabla_\theta f(x'; \theta_t) \rangle$$

with the associated time-dependent operator $T_n^t$

$$T_n^t g(x) := \frac{1}{n} \sum_{i=1}^n K_t(x, x_i)g(x_i) = \int_X K_t(x, s)g(s)d\widehat{\rho}(s). \tag{4}$$

The update rule for the residual $r_t$ under gradient flow is given by

$$\partial_t r_t(x) = -\frac{1}{n} \sum_{i=1}^n K_t(x, x_i)r_t(x_i) = -T_n^t r_t.$$

Speaking loosely, as the network width tends to infinity the time-dependent NTK $K_t(x, x')$ becomes constant so that $K_t(x, x') = K^\infty(x, x')$ uniformly in $t$. If $K_t = K^\infty$ then we have the operator equality $T_n^t = T_n$. Similarly, heuristically as $n \to \infty$ we have $T_n \to T_{K^\infty}$. Thus in the idealized infinite width, infinite data limit the update rule becomes

$$\partial_t r_t = -T_{K^\infty} r_t,$$

which has the solution $r_t = \exp(-T_{K^\infty} t)r_0$ which is defined via its projections

$$\langle r_t, \phi_i \rangle_\rho = \exp(-\sigma_i t)\langle r_0, \phi_i \rangle_\rho.$$

Thus in this idealized setting the network learns eigenfunctions $\phi_i$ at rates determined by their eigenvalues $\sigma_i$. The dependence of the convergence rate on the magnitude of $\sigma_i$ is particularly relevant as the NTK tends to have a very skewed spectrum. We can estimate the spectrum of $K^\infty$ by randomly initializing a network and computing the Gram matrix $(G_0)_{i,j} := K_0(x_i, x_j)$. In Figure 1 we plot the spectrum of the NTK Gram Matrix $(G_0)_{i,j} := K_0(x_i, x_j)$ at initialization. We observe a small number of outlier eigenvalues of large magnitude followed by a long tail of small eigenvalues. This phenomenon has appeared in many works (e.g. Arora et al. 2019a; Oymak et al. 2020; Li et al. 2020). For ReLU networks the spectrum is known to asymptotically follow a power law $\sigma_i \sim \Lambda i^{-\nu}$ (Velikanov & Yarotsky, 2021). The goal of this work is to quantify the extent to which a finite-width network trained on finitely many samples behaves like the idealized kernel dynamics $r_t = \exp(-T_{K^\infty} t)r_0$ corresponding to infinite width and infinite data.

## 2.3 Applicable Architectures

We now specify an architecture for our model $f(x; \theta)$. We consider deep networks of the form

$$\alpha^{(0)} := x,$$
$$\alpha^{(l)} := \psi_l(\theta^{(l)}, \alpha^{(l-1)}), \quad l \in [L],$$
$$f(x; \theta) := \frac{1}{\sqrt{m_L}} v^T \alpha^{(L)},$$

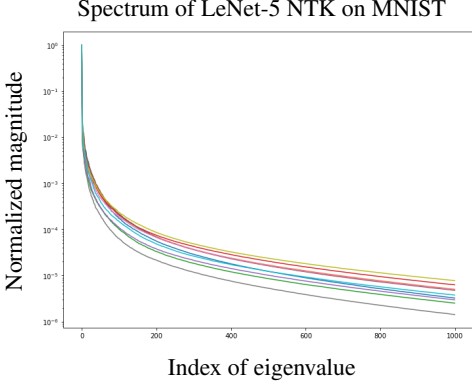
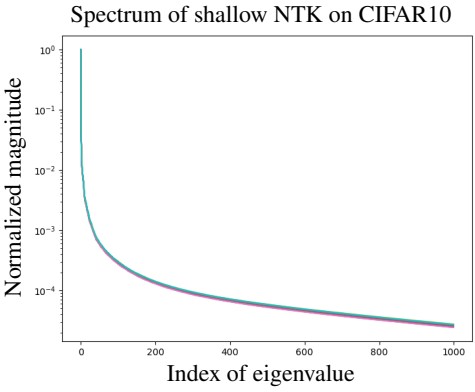

Figure 1: We plot the NTK spectrum on MNIST and CIFAR10 for two networks using 10 random parameter initializations and data batches. In both plots the x-axis represents the eigenvalue index $k$ (linear scale) and the y-axis the normalized eigenvalue $\lambda_k/\lambda_1$ magnitude (log scale). To avoid numerical issues, we compute the NTK on a batch of size 2000 and plot the first 1000 eigenvalues. The left plot computed the NTK corresponding to the logit of class 0 for LeNet-5 on MNIST. The right plot is for a shallow fully-connected softplus network with 4000 hidden units on CIFAR10.

where each $\psi_l(\theta^{(l)}, \bullet) : \mathbb{R}^{m_{l-1}} \to \mathbb{R}^{m_l}$ is a vector-valued function parameterized by $\theta^{(l)} \in \mathbb{R}^{p_l}$ and $v \in \mathbb{R}^{m_L}$. We define $\theta^{(L+1)} := v$ and set $\theta := ((\theta^{(1)})^T, \ldots, (\theta^{(L+1)})^T)^T$ to be the collection of all parameters. We assume each layer mapping $\psi_l$ has one of the following forms:

$$\text{Fully Connected}: \psi_l(\theta^{(l)}, \alpha^{(l-1)}) = \omega\left(\frac{1}{\sqrt{m_{l-1}}} W^{(l)} \alpha^{(l-1)}\right)$$

$$\text{Convolutional}: \psi_l(\theta^{(l)}, \alpha^{(l-1)}) = \omega\left(\frac{1}{\sqrt{m_{l-1}}} W^{(l)} * \alpha^{(l-1)}\right)$$

$$\text{Residual}: \psi_l(\theta^{(l)}, \alpha^{(l-1)}) = \omega\left(\frac{1}{\sqrt{m_{l-1}}} W^{(l)} \alpha^{(l-1)}\right) + \alpha^{(l-1)}$$

Here $\theta^{(l)} = vec(W^{(l)})$ and $\omega$ is a twice continuously differentiable function such that $\omega$ and $\omega'$ are Lipschitz. All parameters of the network will be trained as in practice. For feedforward and residual layers $W^{(l)} \in \mathbb{R}^{m_l \times m_{l-1}}$ is a matrix. For the case of convolutional layers $W^{(l)} \in \mathbb{R}^{K \times m_l \times m_{l-1}}$ is an order-3 tensor with filter size $K$. The precise definition of the convolution $*$ is offered in the appendix. We will let $m = \min_l m_l$ denote the minimum width of the network. We will assume that $\max_l \frac{m_l}{m} = O(1)$. The input dimension $d := m_0$, the depth $L$, and the filter sizes $K$ of convolutional layers will be treated as constant. The depth $L$ being constant is essential for NTK convergence; see Hanin & Nica (2020) for an explanation of failure modes whenever depth is nonconstant.

We will now discuss our initialization scheme. We will perform the antisymmetric initialization trick introduced by Zhang et al. (2020) so that the model is identically zero at initialization $f(\bullet; \theta_0) \equiv 0$. Let $f(x; \theta)$ be any neural network of the form described above. Then let $\tilde{\theta} = \begin{bmatrix} \theta \\ \theta' \end{bmatrix}$ where $\theta, \theta' \in \mathbb{R}^p$. We then define

$$f_{ASI}(x; \tilde{\theta}) := \frac{1}{\sqrt{2}} f(x; \theta) - \frac{1}{\sqrt{2}} f(x; \theta')$$

which takes the difference of two rescaled copies of our original model $f(x; \theta)$ with parameters $\theta$ and $\theta'$ that are optimized freely. The antisymmetric initialization trick initializes $\theta_0 \sim N(0, I)$ then sets $\tilde{\theta}_0 = \begin{bmatrix} \theta_0 \\ \theta_0 \end{bmatrix}$. We then optimize the model $f_{ASI}$ starting from the initialization $\tilde{\theta}_0$. This trick simultaneously ensures that the model is identically zero at initialization without changing the NTK at initialization (Zhang et al., 2020). For ease of notation we will simply assume from now on that $f(x; \theta) = f_{ASI}(x; \theta)$ and not write the subscript $ASI$.

# 3 Main Results

Before stating our main result, we enumerate our key assumptions for the sake of clarity, assumed to hold throughout. Detailed proofs are deferred to the appendix.

**Assumption 3.1.** *The activation $\omega$ is twice continuously differentiable and $\omega$ and $\omega'$ are Lipschitz.*

**Assumption 3.2.** *The input domain $X$ is compact with strictly positive Borel measure $\rho$.*

**Assumption 3.3.** *The target function $f^*$ satisfies $\|f^*\|_{L^\infty(X,\rho)} = O(1)$.*

**Assumption 3.4.** *We use the antisymmetric initilization trick so that $f(\bullet; \theta_0) \equiv 0$.*

Most activation functions except for ReLU satisfy Assumption 3.1, such as Softplus $\omega(x) = \ln(1 + e^x)$, Sigmoid $\omega(x) = \frac{1}{1+e^{-x}}$, and Tanh $\omega(x) = \frac{e^x - e^{-x}}{e^x + e^{-x}}$. Assumption 3.2 is a sufficient condition for Mercer's Theorem to hold. While Mercer's theorem is often assumed to hold implicitly, we prefer to make this assumption explicit. Assumption 3.3 simply means the target function is bounded. We believe the antisymmetric initialization specified in Assumption 3.4 is not strictly necessary but it greatly simplifies the proofs and associated bounds. To sidestep 3.4 one would utilize high probability bounds on the magnitude $|f(x; \theta_0)|$ at initialization. In the following results $f(x; \theta)$ will be any of the architectures discussed in Section 2.3. We are now ready to introduce the main result.

**Theorem 3.5.** *Let $T \geq 1, \epsilon > 0$. Let $K(x, x')$ be a fixed continuous, symmetric, positive definite kernel. For $k \in \mathbb{N}$ let $P_k : L^2(X, \rho) \to L^2(X, \rho)$ denote the orthogonal projection onto the span of the top $k$ eigenfunctions of the operator $T_K$ defined in Equation (2). Let $\sigma_k > 0$ denote the $k$-th eigenvalue of $T_K$. Then $m = \tilde{\Omega}(T^4/\epsilon^2)$ and $n = \tilde{\Omega}(T^2/\epsilon^2)$ suffices to ensure with probability at least $1 - O(mn) \exp(-\Omega(\log^2(m)))$ over the parameter initilization $\theta_0$ and the training samples $x_1, \ldots, x_n$ that for all $t \leq T$ and $k \in \mathbb{N}$*

$$\|P_k(r_t - \exp(-T_K t)r_0)\|_{L^2(X,\rho)}^2 \leq \left[\frac{1 - \exp(-\sigma_k t)}{\sigma_k}\right]^2 \cdot \left[4\|f^*\|_\infty^2 \|K - K_0\|_{L^2(X^2, \rho \otimes \rho)}^2 + \epsilon\right]$$

*and*

$$\|r_t - \exp(-T_K t)r_0\|_{L^2(X,\rho)}^2 \leq t^2 \cdot \left[4\|f^*\|_\infty^2 \|K - K_0\|_{L^2(X^2, \rho \otimes \rho)}^2 + \epsilon\right].$$

## 3.1 Interpretation and Consequences

Theorem 3.5 compares the dynamics of the residual $r_t(x) := f(x; \theta_t) - f^*(x)$ of our finite-width model trained on finitely many samples to the idealized dynamics of a kernel method $\exp(-T_K t)r_0$ with infinite data. We recall that if $\phi_i$ is an eigenfunction of $T_K$ with eigenvalue $\sigma_i$ then $\langle \exp(-T_K t)r_0, \phi_i \rangle_\rho = \exp(-\sigma_i t)\langle r_0, \phi_i \rangle_\rho$. Thus the term $\exp(-T_K t)r_0$ learns the projection along eigenfunction $\phi_i$ linearly at rate $\sigma_i$. Whenever the NTK at initialization $K_0$ concentrates around $K$, the residual $r_t$ will inherit this bias of the kernel dynamics $\exp(-T_K t)r_0$. Furthermore, the bound for the projected difference $\|P_k(r_t - \exp(-T_K t)r_0)\|_{L^2(X,\rho)}^2$ is smaller whenever $\sigma_k$ is large. Therefore the bias appears more pronounced along eigendirections with large eigenvalues.

**Consequences for the special case $K = K^\infty$** In the infinite width limit, we have that $K_0$ approaches $K^\infty$ for general architectures (Yang, 2020). For fixed $x, x'$, by concentration results the typical rate of convergence is $|K_0(x, x') - K^\infty(x, x')| = \tilde{O}(1/\sqrt{m})$ with high probability (Du et al., 2019b,a; Huang & Yau, 2020). Bounds that hold uniformly over $x, x'$ of the same rate were provided by Bowman & Montúfar (2022) and Buchanan et al. (2021). A more pessimistic estimate of $1/m^{1/4}$ is provided by Arora et al. (2019b). Even if the rate is $1/m^{1/4}$, we have that $m = \tilde{\Omega}(\epsilon^{-2})$ is strong enough to ensure that $|K_0(x, x') - K^\infty(x, x')| \leq \epsilon^{1/2}$. Given these results, it is reasonable to make the following assumption for the architectures we consider (see Appendix E).

**Assumption 3.6.** *$m = \tilde{\Omega}(\epsilon^{-2})$ suffices to ensure that $\|K_0 - K^\infty\|_{L^2(X \times X, \rho \otimes \rho)}^2 \leq \epsilon$ holds with high probability $1 - \delta(m)$ over the initialization $\theta_0$ where $\delta(m) = o(1)$.*

Under this assumption, by setting $K = K^\infty$ in Theorem 3.5 we get the following corollary.

**Corollary 3.7.** *Let $\delta(m)$ be defined as in Assumption 3.6 which we assume to hold. Let $T \geq 1$ and $\epsilon > 0$. For $k \in \mathbb{N}$ let $P_k : L^2(X, \rho) \to L^2(X, \rho)$ denote the orthogonal projection onto the span of*

*the top $k$ eigenfunctions of the operator $T_{K^\infty}$ defined in Equation (1). Let $\sigma_k > 0$ denote the $k$-th eigenvalue of $T_{K^\infty}$. Then $m = \tilde{\Omega}(T^4/\epsilon^2)$ and $n = \tilde{\Omega}(T^2/\epsilon^2)$ suffices to ensure with probability at least $1 - O(mn)\exp(-\Omega(\log^2(m)) - \delta(m)$ that for all $t \le T$ and $k \in \mathbb{N}$*

$$\|P_k(r_t - \exp(-T_{K^\infty}t)r_0)\|^2_{L^2(X,\rho)} \le \left[\frac{1 - \exp(-\sigma_k t)}{\sigma_k}\right]^2 \cdot \epsilon$$

*and*

$$\|r_t - \exp(-T_{K^\infty}t)r_0\|^2_{L^2(X,\rho)} \le t^2 \cdot \epsilon.$$

Informally Corollary 3.7 states that up to the stopping time $T$, we have that $r_t \approx \exp(-T_{K^\infty}t)r_0$. As discussed before, the term $\exp(-T_{K^\infty}t)r_0$ projected along the $i$-th eigenfunction of $K^\infty$ decays linearly, $\langle\exp(-T_{K^\infty}t)r_0, \phi_i\rangle_\rho = \exp(-\sigma_i t)\langle r_0, \phi_i\rangle_\rho$. Given that $K^\infty$ tends to have a highly skewed spectrum (see, e.g. Figure 1), the effect the magnitude of $\sigma_i$ has on the convergence rate is particularly relevant. Furthermore the bound on the projected difference $\|P_k(r_t - \exp(-T_{K^\infty}t)r_0)\|_{L^2(X,\rho)}$ is smaller whenever $\sigma_k$ is large due to the dependence of the bound on the inverse eigenvalue $\sigma_k^{-1}$. Thus we have that the bias along the top eigenfunctions is particularly pronounced. Hence we make the following important observation.

**Observation 3.8.** *At the beginning of training the network learns projections along eigenfunctions of the Neural Tangent Kernel integral operator $T_{K^\infty}$ at rates corresponding to their eigenvalues. This is particularly true for the eigenfunctions with large eigenvalues.*

**Scaling with respect to width and number of training data samples** Now let us interpret how the width $m$ and number of training samples $n$ in the theorem scale. We note that as long as $n \le m^\alpha$ for some $\alpha > 0$ the failure probability $O(mn)\exp(-\Omega(\log^2(m)))$ goes to zero as $m \to \infty$. Thus once $m$ and $n$ are sufficiently large relative to the stopping time $T$ and precision $\epsilon$, they can both tend to infinity at just about any rate to achieve a high probability bound. We also observe that $m$ and $n$ both have the same scaling with respect to $\epsilon$, namely $m, n = \tilde{\Omega}(\epsilon^{-2})$. Thus for a fixed stopping time $T$ we can send $m$ and $n$ to infinity at the same rate $m \sim n$ to send the error $\epsilon \to 0$. This is significant as typical NTK analysis requires $m = \Omega(poly(n))$. We reach following important conclusion.

**Observation 3.9.** *The network will inherit the bias of the kernel at the beginning of training even when the width $m$ only grows linearly with the number of samples $n$.*

**Scaling with respect to stopping time** We will now address the scaling with respect to the stopping time $T$. The relevant question is how quickly the terms $P_k\exp(-T_{K^\infty}t)r_0$ and $\exp(-T_{K^\infty}t)r_0$ converge to zero. We observe that

$$\|P_k\exp(-T_{K^\infty}t)r_0\|_{L^2(X,\rho)} \le \exp(-\sigma_k t)\|r_0\|_{L^2(X,\rho)} \le \exp(-\sigma_k t)\|f^*\|_{L^\infty(X,\rho)},$$

where we have used the antisymmetric initialization $r_0 = f(\bullet; \theta_0) - f^* = 0 - f^* = -f^*$ and the basic inequality $\|\bullet\|_{L^2(X,\rho)} \le \|\bullet\|_{L^\infty(X,\rho)}$. Based on this we have that $t \ge \log(\|f^*\|_{L^\infty(X,\rho)}/\epsilon)/\sigma_k$ suffices to ensure $\|P_k\exp(-T_{K^\infty}t)r_0\|_{L^2(X,\rho)} \le \epsilon$. Using this fact we get the following corollary.

**Corollary 3.10.** *Let $\delta(m)$ be defined as in Assumption 3.6 which is assumed to hold. Let $T = \tilde{\Omega}(1/\sigma_k)$ and $\epsilon > 0$. For $k \in \mathbb{N}$ let $P_k : L^2(X, \rho) \to L^2(X, \rho)$ denote the orthogonal projection onto the span of the top $k$ eigenfunctions of the operator $T_{K^\infty}$ defined in Equation (1). Let $\sigma_k > 0$ denote the $k$-th eigenvalue of $T_{K^\infty}$. Then $m = \tilde{\Omega}(\sigma_k^{-8}/\epsilon^2)$ and $n = \tilde{\Omega}(\sigma_k^{-6}/\epsilon^2)$ suffices to ensure that with probability at least $1 - O(mn)\exp(-\Omega(\log^2(m)) - \delta(m)$*

$$\|P_k r_T\|^2_{L^2(X,\rho)} \le \epsilon$$

*and in particular*

$$\frac{1}{2}\|r_T\|^2_{L^2(X,\rho)} \le \tilde{O}(\epsilon) + \|(I - P_k)r_0\|^2_{L^2(X,\rho)}.$$

The interpretation of the Corollary 3.10 is that the stopping time $T = \tilde{\Omega}(1/\sigma_k)$ is long enough to ensure that the network has learned the top $k$ eigenfunctions to $\epsilon$ accuracy provided that $m = \tilde{\Omega}(\sigma_k^{-8}\epsilon^{-2})$ and $n = \tilde{\Omega}(\sigma_k^{-6}\epsilon^{-2})$. We note that the second conclusion of Corollary 3.10 is a bound on the test error $\frac{1}{2}\|r_t\|^2_{L^2(X,\rho)}$. From the antisymmetric initialization $r_0 = -f^*$ so that

$\|(I - P_k)r_0\|_{L^2(X,\rho)}^2 = \|(I - P_k)f^*\|_{L^2(X,\rho)}^2$. For a general target $f^*$, this quantity can decay arbitrary slowly with respect to $k$. Our goal with Theorem 3.5 was not to get a learning guarantee, but to describe how the bias of the kernel $K^\infty$ is inherited by the finite-width network at the beginning of training even for general target functions. Nevertheless we will briefly sketch how it is possible to get a learning guarantee from Corollary 3.7 when $f^*$ is in the RKHS of $K^\infty$. In this case one can show that $\|\exp(-T_{K^\infty}t)r_0\|_{L^2(X,\rho)}^2 = O\left(\frac{\|f^*\|_{\mathcal{H}}^2}{t}\right)$ where $\|\bullet\|_{\mathcal{H}}$ is the RKHS norm. Then treating $\|f^*\|_{\mathcal{H}}$ as a constant one can choose the stopping time $T \sim \epsilon^{-1}$ to bring the test error to $\epsilon$ provided that $m, n = \tilde{\Omega}(poly(\epsilon^{-1}))$. More generally Velikanov & Yarotsky (2021) derive sufficient conditions for the power law $\|\exp(-T_{K^\infty}t)r_0\|_{L^2(X,\rho)}^2 \sim Ct^{-\xi}$ to hold. Using a similar argument in this case one can choose the stopping time $T \sim \epsilon^{-1/\xi}$ and get a learning guarantee for $m, n = \tilde{\Omega}(poly(\epsilon^{-1}))$.

## 3.2 Technical Comparison to Prior Work

Lee et al. (2019); Arora et al. (2019b) compared the network $f(x; \theta)$ to its linearization $f_{lin}(x; \theta) := \langle \nabla_\theta f(x; \theta_0), \theta - \theta_0 \rangle + f(x; \theta_0)$ in the regime where $m = \Omega(poly(n))$. When $m = \Omega(poly(n))$ one can show the loss converges to zero and the parameter changes $\|\theta_t - \theta_0\|_2$ are bounded. By contrast we avoid the condition $m = \Omega(poly(n))$ by employing a stopping time. Arora et al. (2019a); Cao et al. (2021); Basri et al. (2020) proved statements similar to Theorem 3.5 and Corollary 3.7 that roughly correspond to replacing $T_{K^\infty}$ with its Gram matrix induced by the training data $(G^\infty)_{i,j} = K^\infty(x_i, x_j)$ and replacing $\rho$ with the empirical measure $\hat{\rho} = \frac{1}{n}\sum_{i=1}^n \delta_{x_i}$. Arora et al. (2019a); Basri et al. (2020) operate in the regime where $m = \Omega(poly(n))$ and as a benefit do not need to employ a stopping time. Cao et al. (2021) instead of requiring $m = \Omega(poly(n))$ requires that the width $m$ satisfies at least $m = \Omega(\max\{\sigma_k^{-14}, \epsilon^{-6}\})$ where $\sigma_k$ is the cutoff eigenvalue. The most similar work is Bowman & Montúfar (2022), which demonstrated a version of Corollary 3.7 for a shallow feedforward network that is underparameterized. If $p$ is the total number of parameters, they require $m = \tilde{\Omega}(\epsilon^{-1}T^2)$ and $n = \tilde{\Omega}(\epsilon^{-1}pT^2)$. This requires the network to be greatly underparameterized $n \gg p$. Our result was able to remove the dependence of $n$ on $p$ and demonstrate the result for general deep architectures at the expense of slightly worse scaling with respect to $T$ and $\epsilon$.

# 4 Proof Sketch

For simplicity we will go through the case where $K = K^\infty$. At a high level the proof revolves around bounding the difference between the operators $T_{K^\infty}$ and $T_n^t$ defined in Equations (1) and (4).

**Bounding Operator Deviations** Bowman & Montúfar (2022) demonstrated

$$r_t = \exp(-T_{K^\infty}t)r_0 + \int_0^t \exp(-T_{K^\infty}(t-s))(T_{K^\infty} - T_n^s)r_s ds.$$

This exhibits the residual $r_t$ as a sum of $\exp(-T_{K^\infty}t)r_0$ and a correction term. The proof of Theorem 3.5 revolves around bounding the correction term which involves bounding

$$\|(T_{K^\infty} - T_n^s)r_s\|_{L^2(X,\rho)} \leq \|(T_{K^\infty} - T_n)r_s\|_{L^2(X,\rho)} + \|(T_n - T_n^s)r_s\|_{L^2(X,\rho)}.$$

At a high level $\|(T_n - T_n^s)r_s\|_{L^2(X,\rho)}$ will be small whenever the kernel deviations $K_0 - K_s$ are small. On the other hand by metric entropy based arguments we have that $\|(T_{K^\infty} - T_n)r_s\|_{L^2(X,\rho)}$ will be small whenever $n$ is large enough relative to the complexity of the residual functions $r_s$.

**Comparison with Linearization** Let $H(x; \theta) := \nabla_\theta^2 f(x; \theta)$ denote the Hessian of our network with respect to the parameters $\theta$ for a fixed input $x$. It turns out that if $\|H(x, \theta)\|_{op}$ was uniformly small over $x$ and $\theta$ then the kernel deviations $K_0 - K_s$ would be bounded and the complexity of our model $f(x; \theta)$ would be controlled by the complexity of the linearized model $f_{lin}(x; \theta) := \langle \nabla_\theta f(x; \theta_0), \theta - \theta_0 \rangle$. The caveat to this approach is we do not in fact have a way to bound the Hessian $H(x, \theta)$ uniformly. However Liu et al. (2020b) demonstrated that for *fixed $x$* and $R > 0$ we have with high probability over the initialization $\theta_0$

$$\sup_{\theta \in \overline{B}(\theta_0, R)} \|H(x, \theta)\|_{op} = \tilde{O}\left(\frac{R}{\sqrt{m}}poly(R/\sqrt{m})\right). \tag{5}$$

Using a priori parameter norm deviation bounds we have that $\|\theta_t - \theta_0\|_2 = O(\sqrt{t})$ and thus we can set $R = O(\sqrt{T})$. The difficulty then arises to get bounds that only depend on the Hessian $H(x; \theta)$ evaluated only on finitely many inputs $x$. We overcome this difficulty by showing for fixed $\theta_0$ one has high probability bounds over the sampling of the training data $x_1, \ldots, x_n$ that only require the Hessian evaluated on a finite point set. This requires some elaborate calculations involving Rademacher complexity. We then use the Fubini-Tonelli theorem and the Hessian bound (5) to get a bound over the simultaneous sampling of $\theta_0$ and $x_1, \ldots, x_n$.

**Covering Number of the Linearized Model**    The complexity of the residual functions $r_s$ up to the stopping time $T$ can be controlled by bounding the complexity of the function class $\mathcal{C} = \{f_{lin}(x; \theta) : \theta \in \overline{B}(\theta_0, R)\}$. In Appendix A we show that the $L^2(X, \rho)$ metric entropy of the linearized model $\mathcal{C} = \{f_{lin}(x; \theta) : \theta \in \overline{B}(\theta_0, R)\}$ is determined by the spectrum of the Fisher Information Matrix

$$F := \int_X \nabla_\theta f(x; \theta_0) \nabla_\theta f(x; \theta_0)^T d\rho(x). \tag{6}$$

Let $\lambda_1^{1/2} \geq \lambda_2^{1/2} \geq \cdots \geq 0$ denote the eigenvalues of $F^{1/2}$. We define the effective rank of $F^{1/2}$ at scale $\epsilon$ as

$$\tilde{p}(F^{1/2}, \epsilon) = |\{i : \lambda_i^{1/2} > \epsilon\}|.$$

This measures the number of dimensions within the unit ball whose image under $F^{1/2}$ can be larger than $\epsilon$ in Euclidean norm. In Appendix A we demonstrate that the $\epsilon$ covering number of $\mathcal{C}$ in $L^2(X, \rho)$, denoted $\mathcal{N}(\mathcal{C}, \|\bullet\|_{L^2(X,\rho)}, \epsilon)$, has the bound

$$\log \mathcal{N}(\mathcal{C}, \|\bullet\|_{L^2(X,\rho)}, \epsilon) = \tilde{O}(\tilde{p}(F^{1/2}, 0.75\epsilon/R)).$$

It turns out that for $\|(T_{K^\infty} - T_n)r_s\|_{L^2(X,\rho)}$ to be on the order of $\epsilon$ we merely need $n$ to be large relative to $\tilde{p}(F^{1/2}, 0.75\epsilon/R)$. By contrast Bowman & Montúfar (2022) required that the network was underparameterized so that $n$ was large relative to the total number of parameters $p$. Since $\tilde{p} \ll p$, this is what lets us relax the sample complexity dramatically. In fact for fixed $R$ and $\epsilon$ we have that $\tilde{p} = \tilde{O}(1)$ with high probability as the width grows to infinity whereas $p \to \infty$. Interestingly, the quantity $\tilde{p}$ for the loss Hessian at convergence was used recently to derive analytical PAC-Bayes bounds (Yang et al., 2021). Note for the squared loss the (empirical) FIM[1] can be taken as an approximation to the Hessian, and at a minimizer this approximation becomes exact. Thus these two notions are closely related.

## 5    Conclusion and Future Directions

We provided quantitative bounds measuring the $L^2$ difference in function space between a finite-width network trained on finitely many samples and the corresponding kernel method with infinite width and infinite data. As a consequence, the network will inherit the bias of the kernel at the beginning of training even when the width scales linearly with the number of samples. This bias is not only over the training data but over the entire input space. The key property that allows this is the low-effective-rank property of the Fisher Information Matrix (FIM) at initialization which controls the capacity of the model at the beginning of training. An interesting avenue for future work is to investigate if flat minima manifesting a FIM of low effective rank at the end of training can be related to the behavior of the network on out-of-sample data after training.

**Limitations**    Our framework can only characterize the network's bias up to a stopping time. There is compelling evidence that the kernel adapts to the target function later in training (Baratin et al., 2021; Atanasov et al., 2022), and this falls outside our framework. Accounting for adaptations in the kernel is an important problem that is still being addressed by the theoretical community.

**Broader impacts**    We do not foresee any negative societal impacts of characterizing the spectral bias of neural networks. To the contrary we believe that cataloging the properties that networks are biased towards in a variety of regimes will be essential to developing fair and interpretable artificial intelligence over the long-term.

---

[1]Note that we define $F$ as an expectation over the true input distribution $\rho$. To approximate the Hessian of the empirical loss one must replace $\rho$ with the empirical measure $\hat{\rho}$.

## Acknowledgments and Disclosure of Funding

This project has received funding from UCLA FCDA, the National Science Foundation Division of Mathematical Sciences (NSF DMS-2145630), and from the European Research Council (ERC) under the European Union's Horizon 2020 research and innovation programme (grant agreement nº 757983). The authors would like to thank Yonatan Dukler for sharing code to compute the NTK Gram matrix in PyTorch.

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
