# Appendix

The appendix is organized as follows.

## A   Covering Number for the Linearized Model

Our approach to generalization will be based on metric entropy (see, e.g., Wainwright, 2019), a fundamental tool in learning theory. We recall some basic definitions.

**Definition A.1.** *Let $V$ be a vector space with seminorm $\|\bullet\|$. For a subset $A \subset V$ we say that $B$ is a proper $\epsilon$-covering of $A$ if $B \subset A$ and for all $a \in A$ there exists $b \in B$ such that $\|a - b\| \leq \epsilon$.*

Since we will concern ourselves solely with proper coverings we may remove the adjective "proper" when discussing $\epsilon$-coverings. A closely related notion is the $\epsilon$-covering number.

**Definition A.2.** *Let $V$ be a vector space with seminorm $\|\bullet\|$ and let $A \subset V$. For $\epsilon > 0$ we define the proper $\epsilon$-covering number of $A$, denoted $\mathcal{N}(A, \|\bullet\|, \epsilon)$, by*

$$\mathcal{N}(A, \|\bullet\|, \epsilon) = \min_{N \,:\, N \text{ is proper } \epsilon\text{-covering of } A} |N|.$$

It is also useful to define the covering number of a set $K$ with respect to another set $L$.

**Definition A.3.** *Let $K$ and $L$ be two subsets of a vector space $V$. We define $\mathcal{N}(K, L)$ as the smallest $n \in \mathbb{N}$ such that there exists $v^{(1)}, \dots, v^{(n)} \in K$ satisfying*

$$K \subset \bigcup_{i=1}^{n} (v^{(i)} + L).$$

Now consider a model $f_{lin}(x; \theta)$ that is potentially nonlinear in $x$ but affine in $\theta$. The motivating example is the following NTK model

$$f_{lin}(x; \theta) = f(x; \theta_0) + \langle \nabla_\theta f(x; \theta_0), \theta - \theta_0 \rangle.$$

We will be interested in deriving covering numbers for such classes of functions. Since translation by a fixed function does not change the covering number we will for convenience assume the model is linear in $\theta$. Thus we will consider models of the form

$$f_{lin}(x; \theta) = \langle g(x), \theta \rangle.$$

The function $g$ can be nonlinear and thus $x \mapsto f_{lin}(x; \theta)$ is typically nonlinear. For the NTK model we have $g(x) = \nabla_\theta f(x; \theta_0)$. Let $X$ be our input space and let $\nu$ be some measure on $X$. We consider $L^2(X, \nu)$ where

$$\|h\|_{L^2(X,\nu)}^2 = \int_X |h(x)|^2 d\nu(x).$$

Throughout we will assume that $\|g\|_2 \in L^2(X, \nu)$ i.e. $\int_X \|g\|_2^2 d\nu < \infty$. We will be interested in deriving covering numbers for classes of functions

$$\mathcal{C}_A := \{f_{lin}(x; \theta) : \theta \in A\}$$

where $A \subset \Theta$ is some subset of parameter space $\Theta$. For now we will assume that $\Theta = \mathbb{R}^p$. We observe that

$$\|f_{lin}(\bullet; \theta_1) - f_{lin}(\bullet; \theta_2)\|^2_{L^2(X,\nu)} = \int_X |\langle g(x), \theta_1 - \theta_2 \rangle|^2 d\nu(x)$$

$$= \int_X (\theta_1 - \theta_2)^T g(x) g(x)^T (\theta_1 - \theta_2) d\nu(x) = (\theta_1 - \theta_2)^T \left[ \int_X g(x) g(x)^T d\nu(x) \right] (\theta_1 - \theta_2).$$

Thus of primary importance is the symmetric positive semidefinite matrix $M := \int_X g(x) g(x)^T d\nu$. When $\nu$ is a probability measure and $f_{lin}(x; \theta)$ is the NTK model we have that

$$M = \mathbb{E}_{x \sim \nu} \left[ \nabla_\theta f(x; \theta_0) \nabla_\theta f(x; \theta_0)^T \right]$$

is the (uncentered) gradient covariance matrix, which can be interpreted as the Fisher Information Matrix (FIM) for the squared loss. The two most interesting cases are when $\nu$ is the true input distribution or $\nu = \frac{1}{n} \sum_{i=1}^n \delta_{x_i}$ is the empirical distribution arising from the training samples. In the former case $M$ is the true (uncentered) gradient covariance matrix and in the latter case $M$ is the (uncentered) empirical covariance. For neural networks the FIM tends to have a very skewed spectrum (is approximately low rank), and thus the relations between the spectrum of $M$ and the covering number will be particularly relevant. We will define the seminorm $\|\bullet\|_M$ as

$$\|v\|_M := \sqrt{v^T M v}.$$

The following lemma relates the covering number $\mathcal{N}(\mathcal{C}_A, \|\bullet\|_{L^2(X,\nu)}, \epsilon)$ to $\mathcal{N}(A, \|\bullet\|_M, \epsilon)$.

**Lemma A.4.** *Let $N \subset A \subset \mathbb{R}^p$. Then $N$ is a proper $\epsilon$-covering of $A$ with respect to the seminorm $\|\bullet\|_M$ if and only if $\mathcal{C}_N$ is a proper $\epsilon$-covering of $\mathcal{C}_A$ with respect to the $L^2(X,\nu)$ norm.*

*Proof.* As we argued before we have that

$$\|f_{lin}(\bullet; \theta_1) - f_{lin}(\bullet; \theta_2)\|^2_{L^2(X,\nu)} = (\theta_1 - \theta_2)^T \left[ \int_X g(x) g(x)^T d\nu(x) \right] (\theta_1 - \theta_2)$$

$$= (\theta_1 - \theta_2)^T M (\theta_1 - \theta_2) = \|\theta_1 - \theta_2\|^2_M.$$

For each function in $h \in \mathcal{C}_A$ pick a representative parameter $\hat{\theta}(h) \in A$ so that $h = f_{lin}(\bullet; \hat{\theta}(h))$ (if $M$ is strictly positive definite $\hat{\theta}(h)$ is unique). We can choose the mapping $h \mapsto \hat{\theta}(h)$ so that the image of $\mathcal{C}_N$ under this mapping is $N$. Suppose $N$ is an $\epsilon$-covering for A with respect to $\|\bullet\|_M$. Then for each $\theta \in A$ we can choose $\theta'$ such that $\|\theta - \theta'\|_M \leq \epsilon$. Well then for any $h \in \mathcal{C}_A$ we can consider $\hat{\theta}(h)$ and choose $\theta' \in N$ such that $\epsilon \geq \|\hat{\theta}(h) - \theta'\|_M = \|f_{lin}(\bullet; \hat{\theta}(h)) - f_{lin}(\bullet; \theta')\|_{L^2(X,\nu)}$. It follows that $\mathcal{C}_N$ is an $\epsilon$-covering of $\mathcal{C}_A$. Conversely suppose now that $\mathcal{C}_N$ is an $\epsilon$-covering of $\mathcal{C}_A$ with respect to $\|\bullet\|_{L^2(X,\nu)}$. Well then for any $\theta \in A$ we can consider $f_{lin}(x; \theta)$ and take $h \in \mathcal{C}_N$ such that $\|f_{lin}(\bullet; \theta) - h(\bullet)\|_{L^2(X,\nu)} \leq \epsilon$. However since $h(\bullet) = f_{lin}(\bullet; \hat{\theta}(h))$ we have that $\epsilon \geq \|f_{lin}(\bullet; \theta) - f_{lin}(\bullet; \hat{\theta}(h))\|_2 = \|\theta - \hat{\theta}(h)\|_M$. Thus $\hat{\theta}(\mathcal{C}_N) = N$ is an $\epsilon$-covering for A. $\square$

Thus covering the space $\mathcal{C}_A$ in $L^2(X,\nu)$ reduces to covering a subset of Euclidean space under the seminorm $\|\bullet\|_M$. By a change of coordinates we will assume without loss of generality that $M$ is diagonal. Let $M^{1/2}$ be the square root of $M$ and let $\sigma_1 \geq \cdots \geq \sigma_p \geq 0$ be the eigenvalues of $M^{1/2}$. We note that

$$\{v \in \mathbb{R}^p : \|v\|_M \leq 1\} = \left\{ v \in \mathbb{R}^p : \sum_{i=1}^p \sigma_i^2 v_i^2 \leq 1 \right\}.$$

Thus the unit ball in $\mathbb{R}^p$ determined by $\|\bullet\|_M$ is the ellipsoid with half-axis lengths $\sigma_i^{-1}$ (if $\sigma_i = 0$ we consider the ellipsoid as being infinite along that dimension). For a general vector $a \in \mathbb{R}^p$ with nonnegative entries we define the ellipse

$$E_a := \left\{ v \in \mathbb{R}^p : \sum_{i=1}^p \frac{v_i^2}{a_i^2} \leq 1 \right\}$$

where in the sum if $a_i = 0$ we interpret $\frac{v_i^2}{a_i^2}$ as $0$ if $v_i = 0$ and infinity otherwise. $E_a$ is the ellipse with half-axis lengths $a_1, a_2, \ldots, a_n$. We will also let $B_r^k \subset \mathbb{R}^k$ denote the closed Euclidean ball in dimension $k$ of radius $r$, specifically

$$B_r^k := \{v \in \mathbb{R}^k : \sum_{i=1}^{k} v_i^2 \leq r\}.$$

Our main study will be bounding $\mathcal{N}(A, \|\bullet\|_M, \epsilon)$ when $A = \{\theta \in \mathbb{R}^p : \|\theta\|_2 \leq R\} = B_R^p$. This amounts to covering a Euclidean ball with ellipsoids determined by $\|\bullet\|_M$. Fortunately, there are well established results for coverings involving ellipsoids. Let $\sigma = (\sigma_1, \ldots, \sigma_p)^T$ denote the spectrum of $M^{1/2}$ and let $M^{-1/2}$ denote the pseudo-inverse of $M^{1/2}$. Let $L$ denote the closed unit ball in $\mathbb{R}^p$ under the seminorm $\|\bullet\|_M$. In geometric terms $\mathcal{N}(B_R^p, \|\bullet\|_M, \epsilon) = \mathcal{N}(B_R^p, \epsilon L)$. We claim that up to an application of $M^{1/2}$ or $M^{-1/2}$, covering $B_R^p$ with translates of $\epsilon L$ is equivalent to covering $E_{\frac{R}{\epsilon}\sigma}$ with translates of $B_1^p$. This is formalized in the following lemma.

**Lemma A.5.** *Let* $M \in \mathbb{R}^{p \times p}$ *be a symmetric positive semidefinite matrix and let* $\sigma = (\sigma_1, \ldots, \sigma_p)^T \in \mathbb{R}^p$ *denote the eigenvalues of* $M^{1/2}$. *Then* $\mathcal{N}(B_R^p, \|\bullet\|_M, \epsilon) = \mathcal{N}(E_{\frac{R}{\epsilon}\sigma}, B_1^p)$.

*Proof.* By a change of basis we can assume without loss of generality that $M$ is diagonal. Let $L$ denote the closed unit ball of $\mathbb{R}^p$ under $\|\bullet\|_M$. We note that in geometric terms $\mathcal{N}(B_R^p, \|\bullet\|_M, \epsilon) = \mathcal{N}(B_R^p, \epsilon L)$. Since we can dilate by $1/\epsilon$ we can replace $R$ with $R/\epsilon$ and $\epsilon$ with $1$. Thus for convenience we will assume for now that $\epsilon = 1$. We note that if $v^{(1)}, \ldots, v^{(n)}$ form an $L$ covering of $B_R^p$ as in

$$B_R^p \subset \bigcup_{i=1}^{n} (v^{(i)} + L),$$

then

$$E_{R\sigma} = M^{1/2}(B_R^p) \subset \bigcup_{i=1}^{n} (M^{1/2}v^{(i)} + M^{1/2}(L)) \subset \bigcup_{i=1}^{n} (M^{1/2}v^{(i)} + B_1^p).$$

Thus $M^{1/2}v^{(1)}, \ldots, M^{1/2}v^{(n)}$ forms a $B_1^p$ covering of $E_{R\sigma}$. Conversely suppose $v^{(1)}, \ldots, v^{(n)}$ satisfy

$$E_{R\sigma} \subset \bigcup_{i=1}^{n} (v^{(i)} + B_1^p)$$

and let $P$ be the projection onto $\text{span}\{e_i : \sigma_i \neq 0\}$ where $e_i$ denotes the $i$th standard basis vector. Then

$$P(B_R^p) = M^{-1/2}(E_{R\sigma}) \subset \bigcup_{i=1}^{n} (M^{-1/2}v^{(i)} + M^{-1/2}(B_1^p)) = \bigcup_{i=1}^{n} (M^{-1/2}v^{(i)} + P(L)).$$

However $L$ is infinitely long along the dimensions outside $im(P)$, and thus

$$B_R^p \subset \bigcup_{i=1}^{n} (M^{-1/2}v^{(i)} + L).$$

Thus $M^{-1/2}v^{(1)}, \ldots, M^{-1/2}v^{(n)}$ form an $L$ covering of $B_R^p$. We conclude that $\mathcal{N}(B_R^p, L) = \mathcal{N}(E_{R\sigma}, B_1^p)$. Thus for general $\epsilon > 0$ we have that $\mathcal{N}(B_R^p, \|\bullet\|_M, \epsilon) = \mathcal{N}(B_R^p, \epsilon L) = \mathcal{N}(B_{R/\epsilon}^p, L) = \mathcal{N}(E_{\frac{R}{\epsilon}\sigma}, B_1^p)$. $\square$

We will let $vol(\bullet)$ denote volume in the standard Lebesgue sense. If $a \in \mathbb{R}^p$ is a vector with positive entries we recall that the volume of an ellipsoid $E_a$ is given by the formula

$$vol(E_a) = vol(B_1^p) \prod_{i=1}^{p} a_i.$$

When most of the $a_i$ are very small we have that $E_a$ is very thin and has small volume and thus we expect the covering number to be small. Coverings for ellipsoids are well established with roots in

geometric functional analysis. The following lemma is phrased the same as Theorems 1 and 2 in Dumer (2006). The result dates back to classic results in geometric functional analysis. Specifically a similar result for more general convex bodies is sketched at the end of Chapter 5 in Pisier (1989) which also appeared in Gordon et al. (1987, Proposition 1.7). We don't need the additional generality for our purposes. We will offer the simplest proof needed for our purposes for completeness and clarity.

**Lemma A.6** (Dumer 2006; Pisier 1989; Gordon et al. 1987). *Let $a \in \mathbb{R}^p$ be a vector with nonnegative entries. Let $J = \{i : a_i > 1\}$, $K = \sum_{i \in J} \log(a_i)$, $\gamma \in (0, 1/2)$, and $\mu_\gamma = |\{i : a_i^2 > (1-\gamma)^2\}|$. Then the proper covering number $\mathcal{N}(E_a, B_1^p)$ satisfies*

$$K \leq \log \mathcal{N}(E_a, B_1^p) \leq K + \mu_\gamma \log\left(\frac{3}{\gamma}\right).$$

*Proof.* We first prove the lower bound. Let $J = \{i : a_i > 1\}$, $m = |J|$, and let $P$ be the orthogonal projection onto $\text{span}\{e_i : i \in J\}$ where $e_i$ denotes the standard basis. Suppose $v^{(1)}, \ldots, v^{(n)}$ are the centers of a $B_1^p$ covering of $E_a$, specifically

$$E_a \subset \bigcup_{i=1}^n (v^{(i)} + B_1^p).$$

Well then

$$P(E_a) \subset \bigcup_{i=1}^n P(v^{(i)} + B_1^p) = \bigcup_{i=1}^n (Pv^{(i)} + B_1^m).$$

Well then by the standard volume estimate we get that

$$n \cdot vol(B_1^m) \geq vol\left(\bigcup_{i=1}^n (Pv^{(i)} + B_1^m)\right) \geq vol(P(E_a))$$

and thus

$$n \geq \frac{vol(P(E_a))}{vol(B_1^m)} = \prod_{i \in J} a_i.$$

Now we prove the upper bound. Let $\gamma \in (0, 1/2)$ and let $J_\gamma = \{i : a_i^2 > (1-\gamma)^2\}$, $\mu_\gamma = |J_\gamma|$, and let $P$ be the orthogonal projection onto $\text{span}\{e_i : i \in J_\gamma\}$. We first notice that if $v \in E_a$ we have that $\|(I - P)v\|_2 \leq 1 - \gamma$, indeed because for $v \in E_a$

$$\sum_{i \notin J_\gamma} \frac{v_i^2}{(1-\gamma)^2} \leq \sum_{i \notin J_\gamma} \frac{v_i^2}{a_i^2} \leq 1.$$

Thus if $v^{(1)}, \ldots, v^{(n)}$ are the centers of a proper $B_\gamma^{\mu_\gamma}$ covering of $P(E_a)$ then by the triangle inequality they also induce a proper $B_1^p$ covering of $E_a$. Thus let $v^{(1)}, \ldots, v^{(n)}$ be a maximal subset of $P(E_a)$ such that for $i \neq j$ $\|v^{(i)} - v^{(j)}\|_2 > \gamma$. By maximality $v^{(1)}, \ldots, v^{(n)}$ form a $B_\gamma^{\mu_\gamma}$ covering of $P(E_a)$. Well then the balls $v^{(i)} + B_{\gamma/2}^{\mu_\gamma}$ are all disjoint and contained in $P(E_a) + B_{\gamma/2}^{\mu_\gamma}$. Thus by the volume estimates

$$n \cdot vol(B_{\gamma/2}^{\mu_\gamma}) = vol\left(\bigcup_{i=1}^n (v^{(i)} + B_{\gamma/2}^{\mu_\gamma})\right) \leq vol\left(P(E_a) + B_{\gamma/2}^{\mu_\gamma}\right).$$

Thus

$$n \leq \frac{vol\left(P(E_a) + B_{\gamma/2}^{\mu_\gamma}\right)}{vol(B_{\gamma/2}^{\mu_\gamma})}.$$

Note that $B_{1-\gamma}^{\mu_\gamma} \subset P(E_a)$ and thus $B_{\gamma/2}^{\mu_\gamma} \subset \frac{\gamma}{2(1-\gamma)} P(E_a)$. Now let $\|\bullet\|_{P(E_a)}$ be the norm on $\mathbb{R}^{\mu_\gamma}$ such that $P(E_a)$ is the unit ball. Then note for $v, w$ such that $v \in P(E_a)$ and $w \in B_{\gamma/2}^{\mu_\gamma}$ we have that

$$\|v + w\|_{P(E_a)} \leq \|v\|_{P(E_a)} + \|w\|_{P(E_a)} \leq 1 + \frac{\gamma}{2(1-\gamma)}.$$

We conclude that $P(E_a) + B_{\gamma/2}^{\mu_\gamma} \subset \left(1 + \frac{\gamma}{2(1-\gamma)}\right) P(E_a)$. Therefore

$$n \leq \frac{vol\left(P(E_a) + B_{\gamma/2}^{\mu_\gamma}\right)}{vol(B_{\gamma/2}^{\mu_\gamma})} \leq \frac{vol\left[\left(1 + \frac{\gamma}{2(1-\gamma)}\right) P(E_a)\right]}{vol(B_{\gamma/2}^{\mu_\gamma})} = \left(\frac{2}{\gamma} + \frac{1}{1-\gamma}\right)^{\mu_\gamma} \prod_{i \in J_\gamma} a_i.$$

Note that since $\gamma < 1/2$ we have that $\frac{1}{1-\gamma} < \frac{1}{\gamma}$. Therefore $\frac{2}{\gamma} + \frac{1}{1-\gamma} \leq \frac{3}{\gamma}$. Moreover $\prod_{i \in J_\gamma} a_i \leq \prod_{i \in J} a_i$. Thus

$$n \leq \left(\frac{2}{\gamma} + \frac{1}{1-\gamma}\right)^{\mu_\gamma} \prod_{i \in J_\gamma} a_i \leq \left(\frac{3}{\gamma}\right)^{\mu_\gamma} \prod_{i \in J} a_i.$$

After taking logarithms we get the desired result. $\qquad\square$

From the Lemmas A.5 and A.6 we see that the covering number $\mathcal{N}(B_R^p, \|\bullet\|_M, \epsilon)$ will depend on how many eigenvalues of $M$ lie above a certain threshold. Let $A \in \mathbb{R}^p$ be a symmetric positive semidefinite square matrix with eigenvalues $\lambda_1 \geq \lambda_2 \geq \cdots \geq \lambda_p \geq 0$. We define the effective rank of $A$ at scale $\epsilon$ as

$$\tilde{p}(A, \epsilon) = |\{i : \lambda_i > \epsilon\}|.$$

This measures the number of dimensions within $B_1$ whose image under $A$ can be larger than $\epsilon$ in Euclidean norm. We will also define

$$|A|_{>c} = \prod_{i:\lambda_i>c} \lambda_i,$$

which can be thought of the determinant of $A$ after removing some eigenvalues. We then have our main result.

**Theorem A.7.** *Let $g : X \to \mathbb{R}^p$ such that $\|g\|_2 \in L^2(X, \nu)$. Let $\mathcal{C} = \{x \mapsto \langle g(x), \theta\rangle : \|\theta\|_2 \leq R\}$, $\gamma \in (0, 1/2)$. Define $M \in \mathbb{R}^{p \times p}$ by*

$$M = \int_X g(x)g(x)^T d\nu(x).$$

*Then the proper covering number $\mathcal{N}(\mathcal{C}, \|\bullet\|_{L^2(X,\nu)}, \epsilon)$ satisfies*

$$\log\left|\frac{R}{\epsilon}M^{1/2}\right|_{>1} \leq \log \mathcal{N}(\mathcal{C}, \|\bullet\|_{L^2(X,\nu)}, \epsilon) \leq \log\left|\frac{R}{\epsilon}M^{1/2}\right|_{>1} + \tilde{p}\left(\frac{R}{\epsilon}M^{1/2}, (1-\gamma)\right) \log\left(\frac{3}{\gamma}\right).$$

*Proof.* We have by Lemmas A.4 and A.5 that $\mathcal{N}(\mathcal{C}, \|\bullet\|_{L^2(X,\nu)}, \epsilon) = \mathcal{N}(B_R^p, \|\bullet\|_M, \epsilon) = \mathcal{N}(E_{\frac{R}{\epsilon}\sigma}, B_1^p)$ where $\sigma = (\sigma_1, \ldots, \sigma_p)^T \in \mathbb{R}^p$ is the vector of eigenvalues of $M^{1/2}$. Well then by applying Lemma A.6 with $a = \frac{R}{\epsilon}\sigma$ we have that

$$\log\left|\frac{R}{\epsilon}M^{1/2}\right|_{>1} \leq \log \mathcal{N}(E_{\frac{R}{\epsilon}\sigma}, B_1) \leq \log\left|\frac{R}{\epsilon}M^{1/2}\right|_{>1} + \tilde{p}\left(\frac{R}{\epsilon}M^{1/2}, (1-\gamma)\right) \log\left(\frac{3}{\gamma}\right).$$

The desired result thus follows. $\qquad\square$

**Corollary A.8.** *Let $g : X \to \mathbb{R}^p$ such that $\|g\|_2 \in L^2(X, \nu)$. Let $\mathcal{C} = \{x \mapsto \langle g(x), \theta\rangle : \|\theta\|_2 \leq R\}$, $\gamma \in (0, 1/2)$. Define $M \in \mathbb{R}^{p \times p}$ by*

$$M = \int_X g(x)g(x)^T d\nu(x).$$

*Then the proper covering number $\mathcal{N}(\mathcal{C}, \|\bullet\|_{L^2(X,\nu)}, \epsilon)$ satisfies*

$$\log \mathcal{N}(\mathcal{C}, \|\bullet\|_{L^2(X,\nu)}, \epsilon) = \tilde{O}\left(\tilde{p}\left(M^{1/2}, \frac{3\epsilon}{4R}\right)\right).$$

*Proof.* This follows from setting $\gamma = 1/4$ and the fact that

$$\log\left|\frac{R}{\epsilon}M^{1/2}\right| = \log\left(\prod_{\sigma_i > \epsilon/R} \frac{R}{\epsilon}\sigma_i\right)$$

$$\leq \tilde{p}(M^{1/2}, \epsilon/R) \log\left(\frac{R\sigma_1}{\epsilon}\right)$$

$$\leq \tilde{p}\left(M^{1/2}, \frac{3\epsilon}{4R}\right) \log\left(\frac{R\sigma_1}{\epsilon}\right).$$

$\square$

# B    Bounding the Network Hessian and other Technical Items

## B.1    Main Hessian Bound

For a fixed input $x$ we will let $H(x, \theta) := \nabla_\theta^2 f(x; \theta)$ denote the Hessian of the network with respect to the parameters. We will use the following result, which follows from the proof of a result by Liu et al. (2020a, Theorem 3.3), which we state here explicitly for reference.

**Theorem B.1** (Reformulation of Liu et al. 2020a, Theorem 3.3)**.** *Let $f(x; \theta)$ be a general neural network of the form specified in Section 2.3 which can be a fully connected network, CNN, ResNet or a mixture of these types. Let $m$ be the minimum of the hidden layer widths and assume $\max_l \frac{m_l}{m} = O(1)$. Given any fixed $R \geq 1$ and $x \in X$ then with probability at least $1 - Cme^{-c\log^2(m)}$*

$$\sup_{\theta \in \overline{B}(\theta_0, R)} \|H(x, \theta)\|_{op} = \tilde{O}\left(\frac{R}{\sqrt{m}}\left[\max\left\{1, \frac{R}{\sqrt{m}}\right\}\right]^{O(L)}\right).$$

*In particular if $\sqrt{m} \geq R$ then*

$$\sup_{\theta \in \overline{B}(\theta_0, R)} \|H(x, \theta)\|_{op} = \tilde{O}\left(\frac{R}{\sqrt{m}}\right).$$

*The constants $c, C > 0$ depend on the architecture but are independent of the width.*

**Discussion of the statement of Theorem B.1**    We note that our statement of Theorem B.1 is not exactly the same as the result of Liu et al. (2020a, Theorem 3.3). Liu et al. (2020a) do not explicitly write the failure probability and the dependence of the Hessian bound on $R$ in the statement of the theorem. In Theorem B.1 we write the failure probability and dependence on the radius $R$ according to the proof[2] provided by the authors Liu et al. (2020a). We also add the assumption $\max_l \frac{m_l}{m} = O(1)$ to the hypothesis. This assumption is so that the initial weight matrices satisfy $\frac{1}{\sqrt{m}}\|W_0^{(l)}\|_{op} = O(1)$ with high probability (see Lemma B.2). This condition on the initial weight matrices appears in the proof by Liu et al. (2020a). The authors Liu et al. (2020a) do not need to explicitly add this assumption because they perform the proof for the case where all the layers have equal width for simplicity of presentation, while stating that the proof generalizes to the case where the layers do not have equal width.

**Exponential dependence on depth**    We note that under the $\tilde{O}$ notation in Theorem B.1 there are constants that depend exponentially on the network depth $L$. For this reason it is essential that the depth $L$ be treated as constant. We will now briefly explain how the term $\max\{1, R/\sqrt{m}\}^{O(L)}$ arises in the bound in Theorem B.1. For simplicity assume the network is fully connected at each layer (the same form of argument holds for the other cases). Let $\xi(\theta) = \max_l \frac{1}{\sqrt{m}}\|W^{(l)}\|_{op}$. With high probability over the initialization we have that $\xi(\theta_0) = O(1)$ (see Lemma B.2). Furthermore for $\theta$ such that $\|\theta - \theta_0\|_2 \leq R$ we have that $\xi(\theta) \leq \xi(\theta_0) + \frac{R}{\sqrt{m}} = O(\max\{1, R/\sqrt{m}\})$. It turns out that

---

[2]We communicated with the authors to better understand the dependence of the bound on the quantity $R$. Nevertheless we accept full liability for any misinterpretation of their proof.

the features $\alpha^{(l)}$ at each layer $l$ satisfy $\frac{1}{\sqrt{m}}\left\|\alpha^{(l)}\right\|_2 = O(\xi^{O(L)})$. Well for $\theta$ such that $\|\theta - \theta_0\|_2 \le R$ as stated before we have that $\xi(\theta) = O(\max\{1, R/\sqrt{m}\})$. Consequently for such $\theta$ we get that $\frac{1}{\sqrt{m}}\left\|\alpha^{(l)}\right\|_2 = O(\xi^{O(L)}) = O(\max\{1, R/\sqrt{m}\}^{O(L)})$. The Hessian bound inherits dependence on the quantity $O(\max\{1, R/\sqrt{m}\}^{O(L)})$ from its dependence the normalized feature $\frac{1}{\sqrt{m}}\left\|\alpha^{(l)}\right\|_2$ norms.

**Antisymmetric initialization and the Hessian** We will now explain how the antisymmetric initialization trick will not hinder us from bounding the Hessian via Theorem B.1. Let $f(x; \theta)$ denote any model of the form specified in Section 2.3 where $\theta \in \mathbb{R}^p$. Let $\tilde{\theta} = \begin{bmatrix} \theta \\ \theta' \end{bmatrix}$ where $\theta, \theta' \in \mathbb{R}^p$. Recall the antisymmetric initialization trick defines the model

$$f_{ASI}(x; \tilde{\theta}) := \frac{1}{\sqrt{2}} f(x; \theta) - \frac{1}{\sqrt{2}} f(x; \theta')$$

which takes the difference of two rescaled copies of the model $f(x; \bullet)$ with parameters $\theta$ and $\theta'$ that are optimized freely. We then note that the Hessian of $f_{ASI}$ has the block diagonal structure

$$\nabla_{\tilde{\theta}}^2 f_{ASI}(x; \tilde{\theta}) = \frac{1}{\sqrt{2}} \begin{bmatrix} \nabla_\theta^2 f(x; \theta) & 0 \\ 0 & -\nabla_{\theta'}^2 f(x; \theta') \end{bmatrix} = \frac{1}{\sqrt{2}} \begin{bmatrix} H(x, \theta) & 0 \\ 0 & -H(x, \theta') \end{bmatrix}.$$

Well then it is not too hard to show that

$$\left\|\nabla_{\tilde{\theta}}^2 f_{ASI}(x; \tilde{\theta})\right\|_{op} \le \max\left[\|H(x, \theta)\|_{op}, \|H(x, \theta')\|_{op}\right].$$

Now recall that the antisymmetric initialization trick initializes $\theta_0 \sim N(0, I)$ then sets $\tilde{\theta}_0 = \begin{bmatrix} \theta_0 \\ \theta_0 \end{bmatrix}$. Furthermore note that if $\left\|\tilde{\theta} - \tilde{\theta}_0\right\|_2 \le R$ then $\|\theta - \theta_0\|_2 \le R$ and $\|\theta' - \theta_0\|_2 \le R$. Thus if $\theta_0$ is an initialization such that the conclusion of Theorem B.1 holds for the model $f(x; \theta)$ then the same conclusion holds for $f_{ASI}(x; \tilde{\theta})$ with initialization $\tilde{\theta}_0$.

## B.2 Definition of the Convolution Operation

In this subsection we will formally define the convolution operation $*$ introduced in Section 2.3. We use the same convention for the convolution operation as Liu et al. (2020a). A convolutional layer of the network has the form

$$\alpha^{(l)} = \psi_l(\theta^{(l)}, \alpha^{(l-1)}) = \omega\left(\frac{1}{\sqrt{m_{l-1}}} W^{(l)} * \alpha^{(l-1)}\right).$$

Here $W^{(l)} \in \mathbb{R}^{K \times m_l \times m_{l-1}}$ is an order-3 tensor where $K$ denotes the filter size, $m_l$ is the number of output channels, and $m_{l-1}$ is the number of input channels. The input $\alpha^{(l-1)} \in \mathbb{R}^{m_{l-1} \times Q}$ is a matrix with $m_{l-1}$ rows as channels and $Q$ columns as pixels. The output of the layer $\psi_l$ is of size $\mathbb{R}^{m_l \times Q}$. From now on we will drop the superscripts and just denote $W = W^{(l)}$ and $\alpha = \alpha^{(l)}$. The convolution operation is defined as

$$(W * \alpha)_{i,q} = \sum_{k=1}^{K} \sum_{j=1}^{m_{l-1}} W_{k,i,j} \alpha_{j, q+k-\frac{K+1}{2}}.$$

This can be reformulated as follows. For each $k \in [K]$ define the matrices $W^{[k]} := W_{k,i,j}$ and $(\alpha^{[k]})_{j,q} := \alpha_{j, q+k-\frac{K+1}{2}}$. Then the convolution operation can be rewritten as

$$(W * \alpha) = \sum_{k=1}^{K} W^{[k]} \alpha^{[k]}.$$

Under this reformulation the convolutional layer can be rewritten as

$$\psi(W, \alpha) = \omega\left(\sum_{k=1}^{K} \frac{1}{\sqrt{m_{l-1}}} W^{[k]} \alpha^{[k]}\right).$$

By treating each $W^{[k]}$ as if it were a weight matrix in a fully connected layer, the convolutional layers can be treated similarly to fully connected layers. Thus when we refer to weight matrices in the context of a convolutional layer we are referring to the matrices $W^{[k]}$.

### B.3 Technical Lemmas

This section will cover some miscellaneous technical lemmas that will be of significance later. The following lemma bounds the operator norm of the weight matrices at initialization.

**Lemma B.2.** *Let $f(x; \theta)$ be a neural network of the form specified in Section 2.3. Assume $m \geq d$ and $\max_l \frac{m_l}{m} \leq A$. Then with probability at least $1 - C \exp(-cm)$ over the initialization $\theta_0$ each weight matrix $W_0$ at initialization satisfies*

$$\frac{1}{\sqrt{m}} \|W_0\| \leq 2\sqrt{A} + 1.$$

*The constant $C > 0$ depends on the architecture but is independent of the width $m$.*

*Proof.* Fix a weight matrix $W \in \mathbb{R}^{m_l \times m_{l-1}}$ in the model. Following Vershynin (2012, Corollary 5.35) we have with probability at least $1 - 2 \exp(-t^2/2)$ over the initialization

$$\|W_0\|_{op} \leq \sqrt{m_l} + \sqrt{m_{l-1}} + t$$

and thus

$$\frac{1}{\sqrt{m}} \left\| W_0^{(l)} \right\|_{op} \leq \frac{\sqrt{m_l}}{\sqrt{m}} + \frac{\sqrt{m_{l-1}}}{\sqrt{m}} + \frac{t}{\sqrt{m}} \leq 2\sqrt{A} + \frac{t}{\sqrt{m}}.$$

Thus by setting $t = \sqrt{m}$ and taking the union bound over all weight matrices in the model (which depends on the architecture) we get the desired result. $\square$

We now state for reference the following lemma which follows from the proof in (Liu et al., 2020a).

**Lemma B.3.** *Let $R \geq 1$ and let $f(x; \theta)$ be a neural network of the form specified in Section 2.3. If $\theta_0$ is an initialization such that each weight matrix $W_0$ satisfies $\frac{1}{\sqrt{m}} \left\| W_0^{(l)} \right\|_2 = O(1)$ then*

$$\sup_{x \in X} \sup_{\theta \in \overline{B}(\theta_0, R)} \|\nabla_\theta f(x; \theta)\|_2 = O\left( \max \left\{ 1, \frac{R}{\sqrt{m}} \right\}^{O(L)} \right).$$

*In particular if $\sqrt{m} \geq R$ then*

$$\sup_{x \in X} \sup_{\theta \in \overline{B}(\theta_0, R)} \|\nabla_\theta f(x; \theta)\|_2 = O(1).$$

As a consequence of the previous lemma we get the following high probability bound on the gradients norm $\|\nabla_\theta f(x; \theta)\|_2$.

**Lemma B.4.** *Let $R \geq 1$ and let $f(x; \theta)$ be a neural network of the form specified in Section 2.3. Assume that $m \geq d$, $\max_l \frac{m_l}{m} = O(1)$, and $\sqrt{m} \geq R$. Then with probability at least $1 - C \exp(-cm)$ over the initialization $\theta_0$ we have that*

$$\sup_{x \in X} \sup_{\theta \in \overline{B}(\theta_0, R)} \|\nabla_\theta f(x; \theta)\|_2 = O(1).$$

*The constant $C > 0$ depends on the architecture but is independent of the width $m$*

*Proof.* This follows immediately from Lemma B.2 and Lemma B.3. $\square$

The following lemma bounds the kernel deviations $K^\theta - K^{\theta_0}$ in terms of the network Hessian.

**Lemma B.5.** *Let $S = \{z_1, \ldots, z_k\} \subset X$. Let $B = \sup_{x \in X} \sup_{\theta \in \overline{B}(\theta_0, R)} \|\nabla_\theta f(x; \theta)\|$ and let $H_{max} = \max_{z \in S} \sup_{\theta \in \overline{B}(\theta_0, R)} \|H(z, \theta)\|_{op}$. Then for $\theta \in \overline{B}(\theta_0, R)$*

$$\max_{i,j \in [k]} |K^\theta(z_i, z_j) - K^{\theta_0}(z_i, z_j)| \leq 2B H_{max} R.$$

*Proof.* We have that

$$
|K^\theta(z_i, z_j) - K^{\theta_0}(z_i, z_j)|
$$
$$
\leq \|\nabla_\theta f(z_i; \theta)\| \, \|\nabla_\theta f(z_j; \theta) - \nabla_\theta f(z_j; \theta_0)\| + \|\nabla_\theta f(z_i; \theta) - \nabla_\theta f(z_i; \theta_0)\| \, \|\nabla_\theta f(z_j; \theta_0)\|
$$
$$
\leq 2BH_{max}R.
$$

Here we have used the fact that

$$
\|\nabla_\theta f(z_i; \theta) - \nabla_\theta f(z_i; \theta_0)\|_2 = \left\| \int_0^1 H(z_i, s\theta + (1-s)\theta_0)(\theta - \theta_0)ds \right\|_2
$$
$$
\leq \int_0^1 \|H(z_i, s\theta + (1-s)\theta_0)\|_{op} \, \|\theta - \theta_0\|_2 \leq H_{max}R.
$$

$\square$

The following lemma provides a trivial bound on $\|\theta_t - \theta_0\|_2$.

**Lemma B.6.**
$$
\|\theta_t - \theta_0\|_2 \leq \frac{\sqrt{t}}{\sqrt{2}} \|\hat{r}_0\|_{\mathbb{R}^n} \leq \frac{\sqrt{t}}{\sqrt{2}} \|f^*\|_{L^\infty(X,\rho)}.
$$

*Proof.*

$$
\|\theta_t - \theta_0\|_2 \leq \int_0^t \|\partial_s \theta_s\|_2 \, ds = \int_0^t \|\partial_\theta L(\theta_s)\|_2 \, ds \leq \sqrt{t} \left[ \int_0^t \|\partial_\theta L(\theta_s)\|_2^2 \, ds \right]^{1/2}
$$
$$
= \sqrt{t} \left[ \int_0^t -\partial_s L(\theta_s) ds \right]^{1/2} = \sqrt{t} \left[ L(\theta_0) - L(\theta_t) \right]^{1/2} \leq \sqrt{t} \left[ L(\theta_0) \right]^{1/2} = \frac{\sqrt{t}}{\sqrt{2}} \|\hat{r}_0\|_{\mathbb{R}^n}
$$
$$
\leq \frac{\sqrt{t}}{\sqrt{2}} \|f^*\|_{L^\infty(X,\rho)}
$$

where the second inequality above follows from the Cauchy-Schwarz inequality and the final inequality follows from the fact that $\|\hat{r}_0\|_{\mathbb{R}^n} = \|y\|_{\mathbb{R}^n} \leq \|f^*\|_{L^\infty(X,\rho)}$ from the antisymmetric initialization. $\square$

## C Convergence of the Operators

Throughout this section $K(x, x')$ will be a fixed continuous, symmetric, positive definite kernel. We will let $\kappa := \max_{x \in X} K(x, x)$. We note that since $K$ is continuous and $X$ is compact we have that $\kappa < \infty$. We will thus treat $\kappa$ as a constant. We also note that since $K$ is a kernel for any $x, x' \in X$ we have the inequality $K(x, x') \leq \sqrt{K(x, x)} \sqrt{K(x', x')} \leq \kappa$.

We will let $K^\theta(x, x') = \langle \nabla_\theta f(x; \theta), \nabla_\theta f(x'; \theta) \rangle$ denote the NTK for a specific parameter $\theta$. In this section $\theta_0$ will be treated as fixed. We will show that for fixed $\theta_0$ we have bounds on $\|(T_K - T_n^s)r_s\|_{L^2(X,\rho)}$ that hold with high probability over the sampling of $S = (x_1, \dots, x_n)$. By the Fubini-Tonelli theorem this suffices to get bounds that hold with high probability over the parameter initialization $\theta_0 \sim \mu$ and data sampling $S \sim \rho^{\otimes n}$ as long as one makes sure that the appropriate events are measureable on the product space. Fortunately, due to the continuity of $K^\theta(x, x')$ and $H(x, \theta)$ with respect to $x, x'$ and $\theta$ we can avoid such issues and we thus will not address measureability line-by-line.

In this section we will bound $\|(T_K - T_n^s)r_s\|_{L^2(X,\rho)}$ for all $s$ such that $\|\theta_s - \theta_0\|_2 \leq R$. This will be done by bounding $\|(T_K - T_n)r_s\|_{L^2(X,\rho)}$ and $\|(T_n - T_n^s)r_s\|_{L^2(X,\rho)}$ separately. At a high level $\|(T_n - T_n^s)r_s\|_{L^2(X,\rho)}$ will be small whenever $K_0 - K_s$ is small. On the other hand $\|(T_K - T_n)r_s\|_{L^2(X,\rho)}$ will be small whenever $n$ is large enough relative to the complexity of the function class $\{f(x; \theta) : \theta \in \overline{B}(\theta_0, R)\}$. If $\sup_{\theta \in \overline{B}(\theta_0, R)} \|H(x, \theta)\|_2$ was uniformly small over $x$ then the kernel deviations $K_0 - K_s$ would be bounded and the complexity of $\{f(x; \theta) : \theta \in \overline{B}(\theta_0, R)\}$ would be controlled by the complexity of the linearized model $f_{lin}(x; \theta) = \langle \nabla_\theta f(x; \theta_0), \theta - \theta_0 \rangle$.

However, Theorem B.1 only gives us the ability to bound $\|H(x,\theta)\|$ for finitely many values of $x$. For this reason we will need to do somewhat elaborate gymnastics using Rademacher complexity to form estimates that only require the evaluation of $\sup_{\theta\in\overline{B}(\theta_0,R)}\|H(x,\theta)\|$ over finitely many values of $x$.

Let $\mathcal{F}$ denote some family of real valued functions and let $S = (z_1,\ldots,z_k)$ be a finite point set. We define
$$\mathcal{F}_{|S} = \{(g(z_1),\ldots,g(z_k)) : g \in \mathcal{F}\}$$
to be the set of all vectors in $\mathbb{R}^k$ formed by restricting a function in $\mathcal{F}$ to the point set $S$. Now let $\epsilon \in \mathbb{R}^k$ be a vector with entries that are i.i.d. Rademacher random variables, i.e. $\epsilon_i \sim \mathrm{Unif}\{+1,-1\}$. We define the (unnormalized) Rademacher complexity of $\mathcal{F}_{|S}$.

$$URad(\mathcal{F}_{|S}) := \mathbb{E}_\epsilon \sup_{v\in\mathcal{F}_{|S}} \langle v,\epsilon\rangle = \mathbb{E}\sup_{g\in\mathcal{F}}\sum_{i=1}^k \epsilon_i g(x_i).$$

We will use the following classic result, see e.g. Telgarsky (2021, Theorem 13.1)

**Theorem C.1.** *Let $\mathcal{F}$ be given with $g(z) \in [a,b]$ a.s. for all $g \in \mathcal{F}$. Then with probability at least $1-\delta$ over the sampling of $z_1,\ldots,z_n$*

$$\sup_{g\in\mathcal{F}}\left[\mathbb{E}[g(Z)] - \frac{1}{n}\sum_{i=1}^n g(z_i)\right] \le \frac{2}{n}URad(\mathcal{F}_{|S}) + 3(b-a)\sqrt{\frac{\log(2/\delta)}{2n}}.$$

We will also make use of the following lemma which is also classic, see e.g. Telgarsky (2021, Lemma 13.3)

**Lemma C.2.** *Let $\ell : \mathbb{R}^n \to \mathbb{R}^n$ be a vector of univariate $C$-lipschitz functions. Then $URad((\ell\circ\mathcal{F})_{|S}) \le C\cdot URad(\mathcal{F}_{|S})$.*

Using this we will now prove the following technical lemma. For the purpose of this lemma $x_1,\ldots,x_n$ will be treated as fixed and the randomness will be over a ghost sample $S' = (x_1',\ldots,x_n')$.

**Lemma C.3.** *Let $R \ge 1$ and $B = \sup_{x\in X}\sup_{\theta\in\overline{B}(\theta_0,R)}\|\nabla_\theta f(x,\theta)\|_2$. Consider $x_1,\ldots,x_n \in X$ to be fixed. Then let*

$$\mathcal{F} = \{x \mapsto \frac{1}{n}\sum_{i=1}^n |K^\theta(x,x_i) - K^{\theta_0}(x,x_i)|^2 : \theta \in \overline{B}(\theta_0,R)\}.$$

*Let $x_1',\ldots,x_n'$ be sampled i.i.d. from $\rho$. Let $S = (x_1,\ldots,x_n)$ and $S' = \{x_1',\ldots,x_n'\}$ and define*

$$H_{max} := \max_{z\in S\cup S'}\sup_{\theta\in\overline{B}(\theta_0,R)}\|H(z,\theta)\|_{op}$$

*Then with probability at least $1-\delta$ over the sampling of $x_1',\ldots,x_n'$ we have that every $g \in \mathcal{F}$ satisfies*

$$\mathbb{E}_{x\sim\rho}[g(x)] \le 12B^2 H_{max}^2 R^2 + 12B^4\sqrt{\frac{\log(2/\delta)}{2n}}.$$

*Proof.* We note that for $\theta \in \overline{B}(\theta_0,R)$

$$|K^\theta(x,x_i) - K^{\theta_0}(x,x_i)|^2 \le [|K^\theta(x,x_i)| + |K^{\theta_0}(x,x_i)|]^2 \le [2B^2]^2 = 4B^4.$$

Therefore for all $g \in \mathcal{F}$ we have that $g(x) \in [0,4B^4]$ a.s. Then by Theorem C.1 we have with probability at least $1-\delta$ over the sampling of $S' = \{x_1',\ldots,x_n'\}$

$$\sup_{g\in\mathcal{F}}\left[\mathbb{E}_{x\sim\rho}[g(x)] - \frac{1}{n}\sum_{i=1}^n g(x_i')\right] \le \frac{2}{n}URad(\mathcal{F}_{|S'}) + 12B^4\sqrt{\frac{\log(2/\delta)}{2n}}.$$

Then we note that for any $z,z' \in S\cup S'$ we by Lemma B.5 that $\theta \in \overline{B}(\theta_0,R)$ implies

$$|K^\theta(z,z') - K^{\theta_0}(z,z')| \le 2BH_{max}R.$$

It follows that for any member of $\mathcal{F}_{|S\cup S'}$ is bounded in infinity norm by $4B^2 H_{max}^2 R^2$. Thus for any $g \in \mathcal{F}$ we have that

$$\frac{1}{n}\sum_{i=1}^n g(x_i') \leq 4B^2 H_{max}^2 R^2$$

and

$$\frac{1}{n} U Rad(\mathcal{F}_{|S'}) \leq 4B^2 H_{max}^2 R^2.$$

Therefore for any $g \in \mathcal{F}$ we have that

$$\mathbb{E}_{x\sim\rho}[g(x)] \leq \frac{1}{n}\sum_{i=1}^n g(x_i') + \frac{2}{n} U Rad(\mathcal{F}_{|S'}) + 12B^4 \sqrt{\frac{\log(2/\delta)}{2n}}$$

$$\leq 12B^2 H_{max}^2 R^2 + 12B^4 \sqrt{\frac{\log(2/\delta)}{2n}}.$$

$\square$

Using the previous lemma we can now bound $\|(T_n - T_n^t)r_t\|_{L^2(X,\rho)}$.

**Lemma C.4.** *Let $R \geq 1$ and $B = \sup_{x\in X}\sup_{\theta\in\overline{B}(\theta_0,R)}\|\nabla_\theta f(x,\theta)\|_2$. Let $S = (x_1,\ldots,x_n)$ and $S' = (x_1',\ldots,x_n')$ be two independent sequences of i.i.d. samples from $\rho$. Define*

$$H_{max} := \max_{z\in S\cup S'}\sup_{\theta\in\overline{B}(\theta_0,R)}\|H(z,\theta)\|_{op}.$$

*Then with probability at least $1-\delta$ over the sampling of $S$ and $S'$ we have that for any $\theta_t$ such that $\|\theta_t - \theta_0\|_2 \leq R$,*

$$\left\|(T_n - T_n^t)r_t\right\|_{L^2(X,\rho)}^2 \leq 2\|f^*\|_{L^\infty(X,\rho)}^2 \left[\|K - K_0\|_{L^2(X^2,\rho\otimes\rho)}^2 + 12B^2 H_{max}^2 R^2 + \tilde{O}\left(\frac{B^4}{\sqrt{n}}\right)\right].$$

*Proof.* We note that

$$\left|(T_n - T_n^t)r_t(x)\right| = \left|\frac{1}{n}\sum_{i=1}^n [K(x,x_i) - K_t(x,x_i)]r_t(x_i)\right|$$

$$\leq \|\hat{r}_t\|_{\mathbb{R}^n}\left[\frac{1}{n}\sum_{i=1}^n |K(x,x_i) - K_t(x,x_i)|^2\right]^{1/2} \leq \|\hat{r}_0\|_{\mathbb{R}^n}\left[\frac{1}{n}\sum_{i=1}^n |K(x,x_i) - K_t(x,x_i)|^2\right]^{1/2}$$

where we have used the property $\|\hat{r}_t\|_{\mathbb{R}^n} \leq \|\hat{r}_0\|_{\mathbb{R}^n}$ from gradient flow. Well from the inequality $(a+b)^2 \leq 2(a^2+b^2)$ we have that

$$\frac{1}{n}\sum_{i=1}^n |K(x,x_i) - K_t(x,x_i)|^2$$

$$\leq \frac{2}{n}\sum_{i=1}^n |K(x,x_i) - K_0(x,x_i)|^2 + \frac{2}{n}\sum_{i=1}^n |K_0(x,x_i) - K_t(x,x_i)|^2.$$

For conciseness let

$$h_1(x) := \frac{1}{n}\sum_{i=1}^n |K(x,x_i) - K_0(x,x_i)|^2$$

$$h_2^t(x) := \frac{1}{n}\sum_{i=1}^n |K_0(x,x_i) - K_t(x,x_i)|^2.$$

Then by the above we have that

$$\left\|(T_n - T_n^t)r_t\right\|_{L^2(X,\rho)}^2 \leq 2\|\hat{r}_0\|_{\mathbb{R}^n}^2 \left[\mathbb{E}_{x\sim\rho}[h_1(x)] + \mathbb{E}_{x\sim\rho}[h_2^t(x)]\right].$$

Well we note that $|K(x, x')| \le \kappa$ and $|K_0(x, x')| \le B^2$ uniformly over $x, x'$. Now consider the random variables $Z_i := \|K(\bullet, x_i) - K_0(\bullet, x_i)\|^2_{L^2(X,\rho)}$ where the randomness is over the sampling of $x_i$. Then we have that $|Z_i| \le [\kappa + B^2]^2$ a.s. Thus by Hoeffding's inequality we have that

$$\mathbb{P}\left(\frac{1}{n}\sum_{i=1}^{n} Z_i - \mathbb{E}_{x_1 \sim \rho}[Z_1] > s\right) \le \exp\left(\frac{-ns^2}{2[\kappa + B^2]^4}\right).$$

Thus with probability at least $1 - \delta$ over the sampling of $x_1, \ldots, x_n$

$$\frac{1}{n}\sum_{i=1}^{n} Z_i \le \mathbb{E}_{x_1 \sim \rho}[Z_1] + \frac{\sqrt{2}[\kappa + B^2]^2\sqrt{\log(1/\delta)}}{\sqrt{n}}. \tag{7}$$

Now note that

$$\frac{1}{n}\sum_{i=1}^{n} Z_i = \mathbb{E}_{x \sim \rho}[h_1(x)] \qquad \mathbb{E}_{x_1 \sim \rho}[Z_1] = \|K - K_0\|^2_{L^2(X^2, \rho \otimes \rho)}.$$

Thus whenever (7) holds we have that

$$\mathbb{E}_{x \sim \rho}[h_1(x)] \le \|K - K_0\|^2_{L^2(X^2, \rho \otimes \rho)} + \frac{\sqrt{2}[\kappa + B^2]^2\sqrt{\log(1/\delta)}}{\sqrt{n}}$$

$$= \|K - K_0\|^2_{L^2(X^2, \rho \otimes \rho)} + \tilde{O}\left(\frac{B^4}{\sqrt{n}}\right).$$

On the other hand we have by Lemma C.3 for any fixed $x_1, \ldots, x_n$ that with probability $1 - \delta$ over the sampling of $x'_1, \ldots, x'_n$ i.i.d. from $\rho$ we have that for all $\theta \in \overline{B}(\theta_0, R)$

$$\mathbb{E}_{x \sim \rho}\left[\frac{1}{n}\sum_{i=1}^{n}|K^\theta(x, x_i) - K^{\theta_0}(x, x_i)|^2\right] \le 12B^2 H^2_{max}R^2 + 12B^4\sqrt{\frac{\log(2/\delta)}{2n}}. \tag{8}$$

Whenever the above holds we have that for any $\theta_t$ such that $\|\theta_t - \theta_0\|_2 \le R$ we have that

$$\mathbb{E}_{x \sim \rho}[h_2^t(x)] \le 12B^2 H^2_{max}R^2 + 12B^4\sqrt{\frac{\log(2/\delta)}{2n}} = 12B^2 H^2_{max}R^2 + \tilde{O}\left(\frac{B^4}{\sqrt{n}}\right).$$

Thus combining these together we have with probability at least $(1 - \delta)^2 \ge 1 - 2\delta$ over the sampling of $x_1, \ldots, x_n, x'_1, \ldots, x'_n$ that Equations (7) and (8) hold simultaneously for all $\theta \in \overline{B}(\theta_0, R)$. In such a case we have that for all $\theta_t$ such that $\|\theta_t - \theta_0\|_2 \le R$ that

$$\mathbb{E}_{x \sim \rho}[h_1(x)] + \mathbb{E}_{x \sim \rho}[h_2^t(x)] \le \|K - K_0\|^2_{L^2(X^2, \rho \otimes \rho)} + 12B^2 H^2_{max}R^2 + \tilde{O}\left(\frac{B^4}{\sqrt{n}}\right).$$

Well then

$$\left\|(T_n - T_n^t)r_t\right\|^2_{L^2(X,\rho)} \le 2\|\hat{r}_0\|^2_{\mathbb{R}^n}\left[\mathbb{E}_{x \sim \rho}[h_1(x)] + \mathbb{E}_{x \sim \rho}[h_2^t(x)]\right]$$

$$\le 2\|\hat{r}_0\|^2_{\mathbb{R}^n}\left[\|K - K_0\|^2_{L^2(X^2, \rho \otimes \rho)} + 12B^2 H^2_{max}R^2 + \tilde{O}\left(\frac{B^4}{\sqrt{n}}\right)\right]$$

$$\le 2\|f^*\|^2_{L^\infty(X,\rho)}\left[\|K - K_0\|^2_{L^2(X^2, \rho \otimes \rho)} + 12B^2 H^2_{max}R^2 + \tilde{O}\left(\frac{B^4}{\sqrt{n}}\right)\right].$$

In the last line above we have used the fact that $\|\hat{r}_0\|_{\mathbb{R}^n} = \|y\|_{\mathbb{R}^n} \le \|f^*\|_{L^\infty(X,\rho)}$ from the antisymmetric initialization. The desired result follows after replacing $\delta$ with $\delta/2$ in the previous argument. $\square$

From Lemma C.4 we get the following corollary.

**Corollary C.5.** *Let $R \geq 1$, $B = \sup_{x \in X} \sup_{\theta \in \overline{B}(\theta_0, R)} \|\nabla_\theta f(x, \theta)\|_2$. Let $S = (x_1, \ldots, x_n)$ and $S' = (x'_1, \ldots, x'_n)$ be two independent sequences of i.i.d. samples from $\rho$. Define*

$$H_{max} := \max_{z \in S \cup S'} \sup_{\theta \in \overline{B}(\theta_0, R)} \|H(z, \theta)\|_{op}.$$

*Then with probability at least $1 - \delta$ over the sampling of $S$ and $S'$ we have that for any $\theta_t$ such that $\|\theta_t - \theta_0\|_2 \leq R$*

$$\left\|(T_n - T_n^t)r_t\right\|_{L^2(X,\rho)}^2 \leq 2 \|f^*\|_{L^\infty(X,\rho)}^2 \|K - K_0\|_{L^2(X^2, \rho \otimes \rho)}^2 + \epsilon$$

*provided that $B = \tilde{O}(1)$, $H_{max} = \tilde{O}(\epsilon^{1/2}/R)$ and $n = \tilde{\Omega}(\epsilon^{-2})$.*

*Proof.* We have by Lemma C.4 with probability at least $1 - \delta$ over the sampling of $S$, $S'$

$$\left\|(T_n - T_n^t)r_t\right\|_{L^2(X,\rho)}^2 \leq 2 \|f^*\|_{L^\infty(X,\rho)}^2 \left[ \|K - K_0\|_{L^2(X^2, \rho \otimes \rho)}^2 + 12 B^2 H_{max}^2 R^2 + \tilde{O}\left(\frac{B^4}{\sqrt{n}}\right) \right].$$

Thus if $B = \tilde{O}(1)$ then $H_{max} = \tilde{O}(\epsilon^{1/2}/R)$ and $n = \tilde{\Omega}(\epsilon^{-2})$ is sufficient to ensure that

$$\left\|(T_n - T_n^t)r_t\right\|_{L^2(X,\rho)}^2 \leq 2 \|f^*\|_{L^\infty(X,\rho)}^2 \|K - K_0\|_{L^2(X^2, \rho \otimes \rho)}^2 + \epsilon.$$

$\square$

Now we will begin the work to bound $\|(T_K - T_n)r_s\|_{L^2(X,\rho)}$. The following technical lemma bounds the Rademacher complexity of the difference between the network $f(x; \theta)$ and the linearization $f_{lin}(x; \theta) = \langle \nabla_\theta f(x; \theta_0), \theta - \theta_0 \rangle$ in terms of the Hessian norm for finitely many values $z \in X$.

**Lemma C.6.** *Let $R \geq 1$, $\mathcal{F} = \{x \mapsto f(x; \theta) - f_{lin}(x; \theta) : \theta \in \overline{B}(\theta_0, R)\}$, $B = \sup_{x \in X} \sup_{\theta \in \overline{B}(\theta_0, R)} \|\nabla_\theta f(x; \theta)\|$, and let $S = (z_1, \ldots, z_n) \subset X$. Furthermore let*

$$H_{max} := \max_{z \in S} \sup_{\theta \in \overline{B}(\theta_0, R)} \|H(z, \theta)\|_{op}.$$

*Then*

$$\sup_{g \in \mathcal{F}} \|g\|_{L^\infty(X,\rho)} \leq 2BR$$

*and*

$$\sup_{g \in \mathcal{F}} \max_{z \in S} |g(z)| \leq \frac{1}{2} R^2 H_{max}.$$

*In particular*

$$\frac{1}{n} U Rad((\mathcal{F} \cup -\mathcal{F})_{|S}) \leq \frac{1}{2} R^2 H_{max}.$$

*Proof.* We note that

$$|f(x; \theta) - f_{lin}(x; \theta)| \leq |f(x; \theta)| + |f_{lin}(x; \theta)|.$$

Well then using the fact that $f(\bullet; \theta_0) = 0$ from the antisymmetric initialization we get

$$|f(x; \theta)| = |f(x; \theta) - f(x; \theta_0)| = \left| \int_0^1 \langle \nabla_\theta f(x; \theta s + (1-s)\theta_0), \theta - \theta_0 \rangle ds \right|$$

$$\leq \int_0^1 |\langle \nabla_\theta f(x; \theta s + (1-s)\theta_0), \theta - \theta_0 \rangle| \leq B \|\theta - \theta_0\| \leq BR.$$

On the other hand

$$|f_{lin}(x; \theta)| = |\langle \nabla_\theta f(x; \theta_0), \theta - \theta_0 \rangle| \leq \|\nabla_\theta f(x; \theta_0)\|_2 \|\theta - \theta_0\|_2 \leq BR.$$

Thus

$$\sup_{\theta \in \overline{B}(\theta_0, R)} \|f(\bullet; \theta) - f_{lin}(\bullet; \theta)\|_{L^\infty(X,\rho)} \leq 2BR$$

and the first conclusion follows. Furthermore by the Lagrange form of the remainder in Taylor's theorem we have for $z \in S$

$$|f(z; \theta) - f_{lin}(z; \theta)| = \left| (\theta - \theta_0)^T \frac{H(z, \xi)}{2} (\theta - \theta_0) \right| \leq \frac{1}{2} \|\theta - \theta_0\|_2^2 \|H(z, \xi)\|_{op}$$

where $\xi$ is some point on the line between $\theta$ and $\theta_0$. Thus if we set

$$H_{max} := \max_{z \in S} \sup_{\theta \in \overline{B}(\theta_0, R)} \|H(z, \theta)\|_{op}$$

we have that

$$|f(z; \theta) - f_{lin}(z; \theta)| \leq \frac{1}{2} R^2 H_{max}$$

for all $\theta \in \overline{B}(\theta_0, R)$. Therefore $\frac{1}{n} URad((\mathcal{F} \cup -\mathcal{F})_{|S}) \leq \frac{1}{2} R^2 H_{max}$ and the desired result follows. $\square$

We now introduce another technical lemma that provides Rademacher complexity and $L^\infty$ norm bounds for the linear model $x \mapsto \langle \nabla_\theta f(x; \theta_0), \theta \rangle$.

**Lemma C.7.** *Let $R \geq 1$, $\mathcal{F} = \{x \mapsto \langle \nabla_\theta f(x; \theta_0), \theta \rangle : \|\theta\|_2 \leq 2R\}$. Let $B = \sup_{x \in X} \sup_{\theta \in \overline{B}(\theta_0, R)} \|\nabla_\theta f(x; \theta)\|$. Then*

$$\sup_{g \in \mathcal{F}} \|g\|_{L^\infty(X, \rho)} \leq 2BR$$

*and*

$$\frac{1}{n} URad(\mathcal{F}_{|S}) \leq \frac{2BR}{\sqrt{n}}.$$

*Proof.* By Cauchy-Schwarz

$$|\langle \nabla_\theta f(x; \theta_0), \theta \rangle| \leq 2BR$$

and thus $\|g\|_{L^\infty(X, \rho)} \leq 2BR$ for all $g \in \mathcal{F}$. Now let $\epsilon \in \mathbb{R}^n$ be a vector with i.i.d Rademacher entries $\epsilon_i \sim \text{Unif}\{+1, -1\}$. Then as was shown by Bartlett & Mendelson (2003, Lemma 22)

$$\mathbb{E}_\epsilon \left[ \sup_{\theta \in \overline{B}(\theta_0, 2R)} \sum_{i=1}^n \epsilon_i \langle \nabla_\theta f(x_i, \theta_0), \theta \rangle \right] = 2R \mathbb{E}_\epsilon \left\| \sum_{i=1}^n \epsilon_i \nabla_\theta f(x_i; \theta_0) \right\|_2$$

$$\leq 2R \left[ \mathbb{E}_\epsilon \left\| \sum_{i=1}^n \epsilon_i \nabla_\theta f(x_i; \theta_0) \right\|_2^2 \right]^{1/2}$$

$$= 2R \left[ \mathbb{E}_\epsilon \left[ \sum_{1 \leq i, j \leq n} \epsilon_i \epsilon_j \langle \nabla_\theta f(x_i; \theta_0), \nabla_\theta f(x_j; \theta_0) \rangle \right] \right]^{1/2}$$

$$= 2R \sqrt{\sum_{i=1}^n K^{\theta_0}(x_i, x_i)}$$

$$\leq 2RB\sqrt{n}.$$

where the first inequality above is an application of Jensen's inequality. The Rademacher complexity bound then follows from the bound above. $\square$

The following lemma compares the $L^2(X, \rho)$ norm to that of its empirical counterpart $L^2(X, \widehat{\rho})$ for the function classes discussed in Lemmas C.6 and C.7.

**Lemma C.8.** *Let $R \geq 1$, $\mathcal{F}_1 = \{x \mapsto f(x; \theta) - f_{lin}(x; \theta) : \theta \in \overline{B}(\theta_0, R)\}$, $\mathcal{F}_2 = \{x \mapsto \langle \nabla_\theta f(x; \theta_0), \theta \rangle : \|\theta\|_2 \leq 2R\}$, and $B = \sup_{x \in X} \sup_{\theta \in \overline{B}(\theta_0, R)} \|\nabla_\theta f(x; \theta)\|$. Then with probability at least $1 - \delta$ over the sampling of $S = (x_1, \ldots, x_n)$*

$$\sup_{g \in \mathcal{F}_1 \cup \mathcal{F}_2} \left| \|g\|_{L^2(X, \rho)}^2 - \|g\|_{L^2(X, \widehat{\rho})}^2 \right| \leq 4BR^3 H_{max} + \tilde{O}\left( \frac{B^2 R^2}{\sqrt{n}} \right).$$

*where $\hat{\rho} = \frac{1}{n}\sum_{i=1}^{n}\delta_{x_i}$ is the empirical measure induced by $x_1, \ldots, x_n$ and*

$$H_{max} := \max_{z \in S} \sup_{\theta \in \overline{B}(\theta_0, R)} \|H(z, \theta)\|_{op}.$$

*Proof.* Let $\mathcal{F} = \{|g|^2 : g \in \mathcal{F}_1 \cup \mathcal{F}_2\}$. Note that by Lemmas C.6 and C.7 we have that for $g \in \mathcal{F}_1 \cup \mathcal{F}_2$ that $\|g\|_{L^\infty(X,\rho)} \leq 2BR$. Thus every $g \in \mathcal{F}$ satisfies $g(x) \in [0, 4B^2R^2]$ a.s. Well then by Theorem C.1 we have with probability at least $1 - \delta$ over the sampling of $S = (x_1, \ldots, x_n)$ that

$$\sup_{g \in \mathcal{F}} \left[ \mathbb{E}_{x \sim \rho}[g(x)] - \frac{1}{n}\sum_{i=1}^{n} g(x_i) \right] \leq \frac{2}{n} URad(\mathcal{F}_{|S}) + 12B^2R^2\sqrt{\frac{\log(2/\delta)}{2n}}.$$

Well note that $x^2$ is $4BR$ Lipschitz on the interval $[-2BR, 2BR]$. Then by Lemma C.2 we have that

$$URad(\mathcal{F}_{|S}) \leq 4BR \cdot URad((\mathcal{F}_1 \cup \mathcal{F}_2)_{|S}).$$

Well then we have that

$$URad((\mathcal{F}_1 \cup \mathcal{F}_2)_{|S}) \leq URad((\mathcal{F}_1 \cup -\mathcal{F}_1 \cup \mathcal{F}_2)_{|S}) \leq URad((\mathcal{F}_1 \cup -\mathcal{F}_1)_{|S}) + URad((\mathcal{F}_2)_{|S})$$

where we have used the property that if $A, A'$ are vector classes such that $\sup_{u \in A}\langle \epsilon, u \rangle \geq 0$ and $\sup_{u \in A'}\langle \epsilon, u \rangle \geq 0$ for all $\epsilon \in \{1, -1\}^n$ then $URad(A \cup A') \leq URad(A) + URad(A')$. Well by Lemma C.6 we have that

$$\frac{1}{n}URad((\mathcal{F}_1 \cup -\mathcal{F}_1)_{|S}) \leq \frac{1}{2}R^2 H_{max}.$$

On the other hand by Lemma C.7 we have that

$$\frac{1}{n}URad((\mathcal{F}_2)_{|S}) \leq \frac{2BR}{\sqrt{n}}.$$

Therefore combining these two bounds we get that

$$\frac{1}{n}URad((\mathcal{F}_1 \cup \mathcal{F}_2)_{|S}) \leq \frac{1}{2}R^2 H_{max} + \frac{2BR}{\sqrt{n}}$$

and thus

$$\frac{1}{n}URad(\mathcal{F}_{|S}) \leq \frac{4BR}{n} \cdot URad((\mathcal{F}_1 \cup \mathcal{F}_2)_{|S}) \leq 4BR \left[ \frac{1}{2}R^2 H_{max} + \frac{2BR}{\sqrt{n}} \right].$$

Therefore by putting everything together we have that

$$\sup_{g \in \mathcal{F}} \left[ \mathbb{E}_{x \sim \rho}[g(x)] - \frac{1}{n}\sum_{i=1}^{n} g(x_i) \right] \leq 8BR \left[ \frac{1}{2}R^2 H_{max} + \frac{2BR}{\sqrt{n}} \right] + 12B^2R^2\sqrt{\frac{\log(2/\delta)}{2n}}$$

$$= 4BR^3 H_{max} + \frac{16B^2R^2}{\sqrt{n}} + 12B^2R^2\sqrt{\frac{\log(2/\delta)}{2n}}.$$

By repeating the same argument for the class $-\mathcal{F}$ and taking a union bound we have with probability at least $1 - 2\delta$ that

$$\sup_{g \in \mathcal{F}} \left| \mathbb{E}_{x \sim \rho}[g(x)] - \frac{1}{n}\sum_{i=1}^{n} g(x_i) \right| \leq 4BR^3 H_{max} + \frac{16B^2R^2}{\sqrt{n}} + 12B^2R^2\sqrt{\frac{\log(2/\delta)}{2n}}.$$

The above can be reinterpreted as

$$\sup_{g \in \mathcal{F}_1 \cup \mathcal{F}_2} \left| \|g\|_{L^2(X,\rho)}^2 - \|g\|_{L^2(X,\hat{\rho})}^2 \right| \leq 4BR^3 H_{max} + \frac{16B^2R^2}{\sqrt{n}} + 12B^2R^2\sqrt{\frac{\log(2/\delta)}{2n}}$$

$$= 4BR^3 H_{max} + \tilde{O}\left( \frac{B^2R^2}{\sqrt{n}} \right).$$

The desired result then follows from replacing $\delta$ with $\delta/2$ in the previous argument. $\square$

Now we are ready to provide a bound on the quantity $\|(T_K - T_n)r(\bullet; \theta)\|_{L^2(X, \rho)}$ for $\theta$ satisfying $\|\theta - \theta_0\|_2 \leq R$.

**Lemma C.9.** *Let $R \geq 1$ and let $B$ and $H_{max}$ be defined as in Lemma C.8. Let $\mathcal{C} = \{x \mapsto f_{lin}(x; \theta) - f^*(x) : \theta \in \overline{B}(\theta_0, R)\}$. Then there are quantities $\Gamma$ and $\Phi$ such that*

$$\Gamma = \tilde{O}\left(\frac{BR\sqrt{\log(\mathcal{N}(\mathcal{C}, L^2(X, \rho), \epsilon))}}{\sqrt{n}}\right)$$

*and*

$$\Phi = 4BR^3 H_{max} + \tilde{O}\left(\frac{B^2 R^2}{\sqrt{n}}\right)$$

*such that with probability at least $1 - \delta$ over the sampling of $x_1, \ldots, x_n$*

$$\sup_{\theta \in \overline{B}(\theta_0, R)} \|(T_K - T_n)r(\bullet; \theta)\|_{L^2(X, \rho)} \leq \Gamma + \kappa\left[\sqrt{R^4 H_{max}^2 + 2\Phi} + \sqrt{4\epsilon^2 + 2\Phi}\right].$$

*Proof.* We will define $r_{lin}(x; \theta) = f_{lin}(x; \theta) - f^*(x)$. Well then we have that

$$\|(T_K - T_n)r(\bullet; \theta)\|_{L^2(X, \rho)}$$
$$\leq \|(T_K - T_n)r_{lin}(\bullet; \theta)\|_{L^2(X, \rho)} + \|(T_K - T_n)(f - f_{lin})(\bullet; \theta)\|_{L^2(X, \rho)}.$$

Now let $E$ be a proper $\epsilon$-covering of $\mathcal{C} = \{r_{lin}(x; \theta) : \theta \in \overline{B}(\theta_0, R)\}$ with respect to $L^2(X, \rho)$. Furthermore assume $E$ is of minimal cardinality so that $|E| = \mathcal{N}(\mathcal{C}, L^2(X, \rho), \epsilon)$. Then for any $r_{lin}(\bullet; \theta)$ we can choose $\hat{\theta} \in \overline{B}(\theta_0, R)$ so that $r_{lin}(\bullet; \hat{\theta}) \in E$ and

$$\left\|r_{lin}(\bullet; \theta) - r_{lin}(\bullet; \hat{\theta})\right\|_{L^2(X, \rho)} \leq \epsilon.$$

Well then

$$\|(T_K - T_n)r_{lin}(\bullet; \theta)\|_{L^2(X, \rho)}$$
$$\leq \left\|(T_K - T_n)r_{lin}(\bullet; \hat{\theta})\right\|_{L^2(X, \rho)} + \left\|(T_K - T_n)(r_{lin}(\bullet; \theta) - r_{lin}(\bullet; \hat{\theta}))\right\|_{L^2(X, \rho)}.$$

We note that for any $r_{lin}(x; \theta) \in \mathcal{C}$ that

$$|r_{lin}(x; \theta)| \leq |f_{lin}(x; \theta)| + |f^*(x)| = |\langle \nabla_\theta f(x; \theta_0), \theta - \theta_0 \rangle| + |f^*(x)|$$
$$\leq BR + \|f^*\|_{L^\infty(X, \rho)} =: S.$$

To handle the term $\left\|(T_K - T_n)r_{lin}(\bullet; \hat{\theta})\right\|_{L^2(X, \rho)}$, for $g \in E$ we define the random variables $Z_i := g(x_i)K_{x_i} - \mathbb{E}_{x \sim \rho}[g(x)K_x]$ taking values in the separable Hilbert space $\mathcal{H}$ where $\mathcal{H}$ is the RKHS associated with $K$. We note that $(T_n - T_K)g$ is equal to $\frac{1}{n}\sum_{i=1}^n Z_i$. Well then note that $\|g(x)K_x\|_{\mathcal{H}} = |g(x)|\|K_x\|_{\mathcal{H}} \leq \|g\|_{L^\infty(X, \rho)}\sqrt{K(x, x)} \leq S\kappa^{1/2}$ a.s. Well then

$$\|Z_i\|_{\mathcal{H}} \leq \|g(x_i)K_{x_i}\|_{\mathcal{H}} + \|\mathbb{E}_{x \sim \rho}[g(x)K_x]\|_{\mathcal{H}}$$
$$\leq S\kappa^{1/2} + \mathbb{E}_{x \sim \rho}\|g(x)K_x\|_{\mathcal{H}} \leq 2S\kappa^{1/2}.$$

Then using Hoeffding's inequality for random variables taking values in a separable Hilbert space (see Rosasco et al. 2010, Section 2.4) we have

$$\mathbb{P}\left(\left\|\frac{1}{n}\sum_{i=1}^n Z_i\right\|_{\mathcal{H}} > s\right) \leq 2\exp\left(-ns^2/2[2S\kappa^{1/2}]^2\right).$$

Thus by the union bound and the fact that $\frac{1}{n}\sum_{i=1}^n Z_i = (T_n - T_K)g$ we have that

$$\mathbb{P}\left(\max_{g \in E}\|(T_n - T_K)g\|_{\mathcal{H}} > s\right) \leq 2|E|\exp\left(-ns^2/2[2S\kappa^{1/2}]^2\right).$$

By setting

$$s = \frac{2\sqrt{2} \cdot S\kappa^{1/2}\sqrt{\log\left(\frac{2|E|}{\delta}\right)}}{\sqrt{n}} = \tilde{O}\left(\frac{BR\sqrt{\log(\mathcal{N}(\mathcal{C}, L^2(X,\rho),\epsilon))}}{\sqrt{n}}\right)$$

we get that with probability at least $1 - \delta$ over the sampling of $x_1, \ldots, x_n$

$$\max_{g \in E} \|(T_n - T_K)g\|_{\mathcal{H}} \le s$$

and thus from the inequality $\|\bullet\|_{L^2(X,\rho)} \le \sqrt{\sigma_1} \|\bullet\|_{\mathcal{H}}$ we get

$$\max_{g \in E} \|(T_n - T_K)g\|_{L^2(X,\rho)} \le s\sqrt{\sigma_1} \le s\sqrt{\kappa}. \tag{9}$$

On the other hand we must bound

$$\left\|(T_K - T_n)(r_{lin}(\bullet; \theta) - r_{lin}(\bullet; \hat{\theta}))\right\|_{L^2(X,\rho)}$$

and

$$\|(T_K - T_n)(f - f_{lin})(\bullet; \theta)\|_{L^2(X,\rho)} .$$

Well note since $K(\bullet, \bullet) \le \kappa$ pointwise it follows by Cauchy-Schwarz that for any $h$

$$|T_K h(x)| = \left|\int K(x,s)h(s)d\rho(s)\right| \le \kappa \|h\|_{L^2(X,\rho)}$$

and similarly

$$|T_n h(x)| = \left|\int K(x,s)h(s)d\hat{\rho}(s)\right| \le \kappa \|h\|_{L^2(X,\hat{\rho})} .$$

Therefore

$$\|(T_K - T_n)h\|_{L^2(X,\rho)} \le \|(T_K - T_n)h\|_{L^\infty(X,\rho)} \le \|T_K h\|_{L^\infty(X,\rho)} + \|T_n h\|_{L^\infty(X,\rho)}$$
$$\le \kappa[\|h\|_{L^2(X,\rho)} + \|h\|_{L^2(X,\hat{\rho})}].$$

Thus we will bound $r_{lin}(\bullet; \theta) - r_{lin}(\bullet; \hat{\theta})$ and $(f - f_{lin})(\bullet; \theta)$ in $L^2(X, \rho)$ and $L^2(X, \hat{\rho})$. Well since $\theta \in \overline{B}(\theta_0, R)$ we have that $(f - f_{lin})(\bullet; \theta) \in \mathcal{F}_1$ where $\mathcal{F}_1$ is defined as in Lemma C.8. On the other hand we note that $r_{lin}(x; \theta) - r_{lin}(x; \hat{\theta}) = \langle \nabla_\theta f(x; \theta_0), \theta - \hat{\theta} \rangle$. Note that since $\theta, \hat{\theta} \in \overline{B}(\theta_0, R)$ we have that $\|\theta - \hat{\theta}\|_2 \le 2R$. Thus $r_{lin}(\bullet; \theta) - r_{lin}(\bullet; \hat{\theta}) \in \mathcal{F}_2$ where $\mathcal{F}_2$ is defined as in Lemma C.8. Thus by Lemma C.8 separate from the randomness before we have with probability at least $1 - \delta$ over the sampling of $x_1, \ldots, x_n$

$$\sup_{g \in \mathcal{F}_1 \cup \mathcal{F}_2} \left|\|g\|_{L^2(X,\rho)}^2 - \|g\|_{L^2(X,\hat{\rho})}^2\right| \le 4BR^3 H_{max} + \tilde{O}\left(\frac{B^2 R^2}{\sqrt{n}}\right) := \Phi. \tag{10}$$

Well note that by Lemma C.6 we have that for each $i \in [n]$

$$|f(x_i; \theta) - f_{lin}(x_i; \theta)| \le \frac{1}{2}R^2 H_{max}$$

and consequently

$$\|f(\bullet; \theta) - f_{lin}(\bullet; \theta)\|_{L^2(X,\hat{\rho})} \le \frac{1}{2}R^2 H_{max}.$$

On the other hand we had by the selection of $\hat{\theta}$ that

$$\left\|r_{lin}(\bullet; \theta) - r_{lin}(\bullet; \hat{\theta})\right\|_{L^2(X,\rho)} \le \epsilon.$$

Now for conciseness let $h_1 = f(\bullet; \theta) - f_{lin}(\bullet; \theta)$ and $h_2 = r_{lin}(\bullet; \theta) - r_{lin}(\bullet; \hat{\theta})$. Then by (10) we have

$$\|h_1\|_{L^2(X,\rho)}^2 \le \|h_1\|_{L^2(X,\hat{\rho})}^2 + \Phi \le \frac{1}{4}R^4 H_{max}^2 + \Phi$$

and

$$\|h_2\|^2_{L^2(X,\widehat{\rho})} \le \|h_2\|^2_{L^2(X,\rho)} + \Phi \le \epsilon^2 + \Phi.$$

This implies

$$\|h_1\|^2_{L^2(X,\rho)} + \|h_1\|^2_{L^2(X,\widehat{\rho})} \le \frac{1}{2}R^4 H^2_{max} + \Phi$$

$$\|h_2\|^2_{L^2(X,\rho)} + \|h_2\|^2_{L^2(X,\widehat{\rho})} \le 2\epsilon^2 + \Phi.$$

Thus using the inequality $a + b \le \sqrt{2}(a^2 + b^2)^{1/2}$ for $a, b \ge 0$ combined with the previous estimates we have

$$\|h_1\|_{L^2(X,\rho)} + \|h_1\|_{L^2(X,\widehat{\rho})} \le \sqrt{2}\sqrt{\frac{1}{2}R^4 H^2_{max} + \Phi} = \sqrt{R^4 H^2_{max} + 2\Phi}$$

and

$$\|h_2\|_{L^2(X,\rho)} + \|h_2\|_{L^2(X,\widehat{\rho})} \le \sqrt{2}\sqrt{2\epsilon^2 + \Phi} = \sqrt{4\epsilon^2 + 2\Phi}.$$

Thus we have just shown that assuming (10) holds that

$$\|(T_K - T_n)h_1\|_{L^2(X,\rho)} \le \kappa[\|h_1\|_{L^2(X,\rho)} + \|h_1\|_{L^2(X,\widehat{\rho})}] \le \kappa\sqrt{R^4 H^2_{max} + 2\Phi}$$

and

$$\|(T_K - T_n)h_2\|_{L^2(X,\rho)} \le \kappa[\|h_2\|_{L^2(X,\rho)} + \|h_2\|_{L^2(X,\widehat{\rho})}] \le \kappa\sqrt{4\epsilon^2 + 2\Phi}.$$

Then by taking a union bound we can assume with probability at least $1 - 2\delta$ that (9) and (10) hold simultaneously. In which case our previous estimates combine to give us the bound

$$\|(T_K - T_n)r(\bullet; \theta)\|_{L^2(X,\rho)}$$

$$\le \left\|(T_K - T_n)r_{lin}(\bullet; \hat{\theta})\right\|_{L^2(X,\rho)} + \|(T_K - T_n)h_1\|_{L^2(X,\rho)} + \|(T_K - T_n)h_2\|_{L^2(X,\rho)}$$

$$\le s\sqrt{\kappa} + \kappa\left[\sqrt{R^4 H^2_{max} + 2\Phi} + \sqrt{4\epsilon^2 + 2\Phi}\right].$$

We now note that as long as (9) and (10) hold the same argument runs through for any $\theta \in \overline{B}(\theta_0, R)$. Thus with probability at least $1 - 2\delta$

$$\sup_{\theta \in \overline{B}(\theta_0, R)} \|(T_K - T_n)r(\bullet; \theta)\|_{L^2(X,\rho)} \le s\sqrt{\kappa} + \kappa\left[\sqrt{R^4 H^2_{max} + 2\Phi} + \sqrt{4\epsilon^2 + 2\Phi}\right].$$

The desired conclusion follows by setting $\Gamma = s\sqrt{\kappa}$ and replacing $\delta$ with $\delta/2$ in the previous argument. $\qquad\square$

From Lemma C.9 we get the following corollary.

**Corollary C.10.** *Let $R \ge 1$ and*

$$B = \sup_{x \in X} \sup_{\theta \in \overline{B}(\theta_0, R)} \|\nabla_\theta f(x, \theta)\|_2.$$

*Then with probability at least $1 - \delta$ over the sampling of $x_1, \ldots, x_n$ we have that*

$$\sup_{\theta \in \overline{B}(\theta_0, R)} \|(T_K - T_n)r(\bullet; \theta)\|^2_{L^2(X,\rho)} \le \epsilon$$

*provided that $B = \tilde{O}(1)$, $H_{max} = \tilde{O}(\epsilon/R^3)$ and $n = \tilde{\Omega}(R^4/\epsilon^2)$ where the expressions under the $\tilde{O}$ and $\tilde{\Omega}$ notation do not depend on the values $x_1, \ldots, x_n$.*

*Proof.* After substituting $\epsilon^{1/2}$ for $\epsilon$ in Lemma C.9 we have that with probability at least $1 - \delta$ over the sampling of $x_1, \ldots, x_n$

$$\sup_{\theta \in \overline{B}(\theta_0, R)} \|(T_K - T_n)r(\bullet; \theta)\|_{L^2(X,\rho)} \le \Gamma + \kappa\left[\sqrt{R^4 H^2_{max} + 2\Phi} + \sqrt{4\epsilon + 2\Phi}\right]$$

where

$$\Gamma = \tilde{O}\left(\frac{BR\sqrt{\log(\mathcal{N}(\mathcal{C}, L^2(X, \rho), \epsilon^{1/2}))}}{\sqrt{n}}\right),$$

$$\Phi = 4BR^3 H_{max} + \tilde{O}\left(\frac{B^2 R^2}{\sqrt{n}}\right),$$

and

$$\mathcal{C} = \{x \mapsto f_{lin}(x; \theta) - f^*(x) : \theta \in \overline{B}(\theta_0, R)\}.$$

Now define

$$F := \int_X \nabla_\theta f(x; \theta_0) \nabla_\theta f(x; \theta_0)^T d\rho(x).$$

Since translation by a fixed function does not change the covering number we have by Corollary A.8
that

$$\log \mathcal{N}(\mathcal{C}, L^2(X, \rho), \epsilon^{1/2}) = \tilde{O}\left(\tilde{p}\left(F^{1/2}\frac{3\epsilon^{1/2}}{4R}\right)\right) = \tilde{O}\left(\tilde{p}\left(F, \frac{9\epsilon}{16R^2}\right)\right).$$

Well using the fact that $\tilde{p}(A, \epsilon) \leq \frac{Tr(A)}{\epsilon}$ we have that

$$\tilde{p}\left(F, \frac{9\epsilon}{16R^2}\right) \leq \frac{16R^2 Tr(F)}{9\epsilon}.$$

Well we note that

$$Tr(F) = Tr(\mathbb{E}_{x \sim \rho}[\nabla_\theta f(x; \theta_0) \nabla_\theta f(x; \theta_0)^T]) = \mathbb{E}_{x \sim \rho} Tr(\nabla_\theta f(x; \theta_0) \nabla_\theta f(x; \theta_0)^T)$$

$$= \mathbb{E}_{x \sim \rho} \|\nabla_\theta f(x; \theta_0)\|^2 \leq B^2.$$

Therefore assuming $B = \tilde{O}(1)$ we have that

$$\Gamma = \tilde{O}\left(\frac{R\sqrt{\log \mathcal{N}(\mathcal{C}, L^2(X, \rho), \epsilon^{1/2})}}{\sqrt{n}}\right) = \tilde{O}\left(\frac{R^2}{\epsilon^{1/2}\sqrt{n}}\right).$$

Thus $n = \tilde{\Omega}(R^4/\epsilon^2)$ suffices to ensure that $\Gamma = O(\epsilon^{1/2})$. Now we must bound

$$\Phi = 4BR^3 H_{max} + \tilde{O}\left(\frac{B^2 R^2}{\sqrt{n}}\right).$$

We note that whenever $B = \tilde{O}(1)$ we have that $H_{max} = \tilde{O}(\epsilon/R^3)$ and $n = \tilde{\Omega}(R^4/\epsilon^2)$ guarantees
that $\Phi = O(\epsilon)$. Finally we have that $H_{max} = \tilde{O}(\epsilon/R^3) \subset \tilde{O}(\epsilon^{1/2}/R^2)$ suffices to ensure that
$R^4 H_{max}^2 = O(\epsilon)$. Thus given all these conditions are met we have that

$$\Gamma + \kappa\left[\sqrt{R^4 H_{max}^2 + 2\Phi} + \sqrt{4\epsilon + 2\Phi}\right] = O(\epsilon^{1/2}).$$

The desired result then follows from setting the constants under the $\tilde{O}$ and $\tilde{\Omega}$ notation appropriately.
$\square$

The following lemma combines the results in this section to get the ultimate bound on the operator
deviations $T_K - T_n^t$.

**Lemma C.11.** *Let $R \geq 1$ and $\epsilon \in (0, R)$. Let $S = (x_1, \ldots, x_n)$ and $S' = (x'_1, \ldots, x'_n)$ be two
separate i.i.d. samples from $\rho$ and denote*

$$H_{max} := \max_{z \in S \cup S'} \sup_{\theta \in \overline{B}(\theta_0, R)} \|H(z, \theta)\|_{op}$$

$$B := \sup_{x \in X} \sup_{\theta \in \overline{B}(\theta_0, R)} \|\nabla_\theta f(x, \theta)\|_2.$$

*Then with probability at least $1 - \delta$ over the sampling of $S$, $S'$ we have that for any $t$ such that
$\|\theta_t - \theta_0\|_2 \leq R$ that*

$$\left\|(T_K - T_n^t)r_t\right\|_{L^2(X, \rho)}^2 \leq 4\|f^*\|_{L^\infty(X, \rho)}^2 \|K - K_0\|_{L^2(X^2, \rho \otimes \rho)}^2 + \epsilon$$

*provided that $B = \tilde{O}(1)$, $H_{max} = \tilde{O}(\epsilon/R^3)$ and $n = \tilde{\Omega}(R^4/\epsilon^2)$ where the expressions under the $\tilde{O}$
and $\tilde{\Omega}$ notation do not depend on $S$ and $S'$.*

*Proof.* We note that for $\theta_t$ such that $\|\theta_t - \theta_0\|_2 \le R$ that

$$
\begin{aligned}
\left\|(T_K - T_n^t)r_t\right\|_{L^2(X,\rho)}^2 &\le [\|(T_K - T_n)r_t\|_{L^2(X,\rho)} + \left\|(T_n - T_n^t)r_t\right\|_{L^2(X,\rho)}]^2 \\
&\le 2\left\|(T_K - T_n)r_t\right\|_{L^2(X,\rho)}^2 + 2\left\|(T_n - T_n^t)r_t\right\|_{L^2(X,\rho)}^2 \\
&\le 2\sup_{\theta \in \overline{B}(\theta_0,R)} \|(T_K - T_n)r(\bullet;\theta)\|_{L^2(X,\rho)}^2 + 2\left\|(T_n - T_n^t)r_t\right\|_{L^2(X,\rho)}^2 .
\end{aligned}
$$

Well by Corollary C.10 we have with probability at least $1 - \delta$ over the sampling of $x_1, \ldots, x_n$

$$
\sup_{\theta \in \overline{B}(\theta_0,R)} \|(T_K - T_n)r(\bullet;\theta)\|_{L^2(X,\rho)}^2 \le \epsilon
$$

provided that $B = \tilde{O}(1)$, $H_{max} = \tilde{O}(\epsilon/R^3)$ and $n = \tilde{\Omega}(R^4/\epsilon^2)$. This result also does not depend in any way on $S'$. On the other hand by Corollary C.5 separate from the randomness before we have with probability at least $1 - \delta$ over the sampling of $S$ and $S'$ that for any $\theta_t$ such that $\|\theta_t - \theta_0\|_2 \le R$

$$
\left\|(T_n - T_n^t)r_t\right\|_{L^2(X,\rho)}^2 \le 2\|f^*\|_{L^\infty(X,\rho)}^2 \|K - K_0\|_{L^2(X^2,\rho\otimes\rho)}^2 + \epsilon.
$$

provided that $B = \tilde{O}(1)$, $H_{max} = \tilde{O}(\epsilon^{1/2}/R)$ and $n = \tilde{\Omega}(\epsilon^{-2})$. The desired result then follows from taking a union bound and replacing $\delta$ with $\delta/2$ and $\epsilon$ with $\epsilon/4$. $\qquad\square$

# D    Main Result

## D.1    Damped Deviations

In this subsection we will recall some definitions and results from Bowman & Montúfar (2022). The main theorems in Bowman & Montúfar (2022) assume that the network architecture is shallow, however the results we recall in this section do not depend on the architecture. Let $K(x, x')$ be a continuous, symmetric, positive-definite kernel. Recall that $K$ defines the integral operator

$$
T_K g(x) := \int_X K(x, s)g(s)d\rho(s).
$$

Then by Mercer's theorem

$$
K(x, x') = \sum_{i=1}^\infty \sigma_i \phi_i(x)\phi_i(x')
$$

where $\{\phi_i\}_i$ is an orthonormal basis of $L^2(X, \rho)$ and $\{\sigma_i\}_i$ is a nonincreasing sequence of positive values. Each $\phi_i$ is an eigenfunction of $T_K$ with eigenvalue $\sigma_i$, i.e. $T_K\phi_i = \sigma_i\phi_i$. Let $x \mapsto g_s(x)$ be a $L^2(X, \rho)$ function for each $s \in [0, t]$. Assume $s \mapsto \langle \phi_i, g_s \rangle_\rho$ is measureable for each $i$ and $\int_0^t \|g_s\|_{L^2(X,\rho)}^2 ds < \infty$. Then we write

$$
\int_0^t g_s ds
$$

to denote the coordinate-wise integral, meaning that $\int_0^t g_s ds$ is the $L^2(X, \rho)$ function $h$ such that

$$
\langle h, \phi_i \rangle_\rho = \int_0^t \langle g_s, \phi_i \rangle_\rho ds.
$$

With this definition in hand we now recall the following "Damped Deviations" lemma given by Bowman & Montúfar (2022, Lemma 2.4).

**Lemma D.1.** *Let $K(x, x')$ be a continuous, symmetric, positive-definite kernel. Let $[T_K h](\bullet) = \int_X K(\bullet, s)h(s)d\rho(s)$ be the integral operator associated with $K$ and let $[T_n^s h](\bullet) = \frac{1}{n}\sum_{i=1}^n K_s(\bullet, x_i)h(x_i)$ denote the operator associated with the time-dependent NTK $K_s$. Then*

$$
r_t = \exp(-T_K t)r_0 + \int_0^t \exp(-T_K(t - s))(T_K - T_n^s)r_s ds,
$$

*where the equality is in the $L^2(X, \rho)$ sense.*

Furthermore we have the following lemma Bowman & Montúfar (2022, Lemma C.8)

**Lemma D.2.** *Let $K(x, x')$ be a continuous, symmetric, positive-definite kernel with associated operator $T_K h(\bullet) = \int_X K(\bullet, s)h(s)d\rho(s)$. Let $T_n^s h(\bullet) = \frac{1}{n}\sum_{i=1}^n K_s(\bullet, x_i)h(x_i)$ denote the operator associated with the time-dependent NTK. Then*

$$\|P_k(r_t - \exp(-T_K t)r_0)\|_{L^2(X,\rho)} \leq \frac{1 - \exp(-\sigma_k t)}{\sigma_k}\sup_{s \leq t}\|(T_K - T_n^s)r_s\|_{L^2(X,\rho)}.$$

*and*

$$\|r_t - \exp(-T_K t)r_0\|_{L^2(X,\rho)} \leq t \cdot \sup_{s \leq t}\|(T_K - T_n^s)r_s\|_{L^2(X,\rho)}.$$

### D.2 Proof of Theorem 3.5

We are now ready to prove the main result of this paper.

**Theorem 3.5.** *Let $T \geq 1, \epsilon > 0$. Let $K(x, x')$ be a fixed continuous, symmetric, positive definite kernel. For $k \in \mathbb{N}$ let $P_k : L^2(X, \rho) \to L^2(X, \rho)$ denote the orthogonal projection onto the span of the top $k$ eigenfunctions of the operator $T_K$ defined in Equation (2). Let $\sigma_k > 0$ denote the $k$-th eigenvalue of $T_K$. Then $m = \tilde{\Omega}(T^4/\epsilon^2)$ and $n = \tilde{\Omega}(T^2/\epsilon^2)$ suffices to ensure with probability at least $1 - O(mn)\exp(-\Omega(\log^2(m))$ over the parameter initilization $\theta_0$ and the training samples $x_1, \ldots, x_n$ that for all $t \leq T$ and $k \in \mathbb{N}$*

$$\|P_k(r_t - \exp(-T_K t)r_0)\|_{L^2(X,\rho)}^2 \leq \left[\frac{1 - \exp(-\sigma_k t)}{\sigma_k}\right]^2 \cdot \left[4\|f^*\|_\infty^2\|K - K_0\|_{L^2(X^2,\rho\otimes\rho)}^2 + \epsilon\right]$$

*and*

$$\|r_t - \exp(-T_K t)r_0\|_{L^2(X,\rho)}^2 \leq t^2 \cdot \left[4\|f^*\|_\infty^2\|K - K_0\|_{L^2(X^2,\rho\otimes\rho)}^2 + \epsilon\right].$$

*Proof.* Let $\theta_0$ be the parameter initialization and let $S = (x_1, \ldots, x_n)$ and $S' = (x_1', \ldots, x_n')$ be two i.i.d. samples from $\rho$. Furthermore let $1 \leq R \leq \sqrt{m}$. Let $E_1 \subset \mathbb{R}^p \times X^{2n}$ be the set of values $(\theta_0, S, S')$ so that the conclusion of Lemma C.11 holds. Similarly let $E_2$ be the set of values $(\theta_0, S, S')$ satisfying

$$B := \max_{x \in X}\sup_{\theta \in \overline{B}(\theta_0, R)}\|\nabla_\theta f(x; \theta)\|_2 = O(1)$$

and

$$H_{max} := \max_{z \in S \cup S'}\sup_{\theta \in \overline{B}(\theta_0, R)}\|H(z, \theta)\|_{op} = \tilde{O}(\epsilon/R^3)$$

where the expression $O(1)$ above is the bound on $B$ given by Lemma B.4 and the expression $\tilde{O}(\epsilon/R^3)$ is precisely the condition on $H_{max}$ in the conclusion of Lemma C.11. By Lemma C.11 for any fixed $\theta_0$ we have that the conclusion holds with probability at least $1 - \delta$ over the sampling of $S, S'$. Thus for any $\theta_0$ we have that

$$\mathbb{E}_{S,S'}[\mathbb{I}\{(\theta_0, S, S') \in E_1\}] \geq 1 - \delta.$$

It follows then by the Fubini-Tonelli theorem that

$$\mathbb{P}(E_1) = \mathbb{E}_{\theta_0}\mathbb{E}_{S,S'}[\mathbb{I}\{(\theta_0, S, S') \in E_1\}] \geq 1 - \delta.$$

On the other hand by Theorem B.1 and Lemma B.4 combined with a union bound we have that for any fixed $S, S'$ then with probability at least $1 - 2Cmn\exp(-c\log^2(m)) - C\exp(-cm)$ that $H_{max} = \tilde{O}(R/\sqrt{m})$ and $B = O(1)$. Thus if $m = \tilde{\Omega}(R^8/\epsilon^2)$ we ensure that $H_{max} = \tilde{O}(\epsilon/R^3)$. Then by the same Fubini-Tonelli argument as before we get that

$$\mathbb{P}(E_2) = \mathbb{E}_{S,S'}\mathbb{E}_{\theta_0}\mathbb{I}\{(\theta_0, S, S') \in E_2\} \geq 1 - 2Cmn\exp(-c\log^2(m)) - C\exp(-cm).$$

Thus by taking a union bound we have with probability at least $1 - \delta - O(mn)\exp(-\Omega(\log^2(m))$ that the events $E_1$ and $E_2$ both hold simultaneously. This holds for any $\delta$ so we may as well set $\delta = O(mn)\exp(-\Omega(\log^2(m)))$ and absorb it into the other term. Whenever $E_1$ and $E_2$ hold simultaneously we have by Lemma C.11 that for any $\theta_t$ such that $\|\theta_t - \theta_0\|_2 \leq R$

$$\left\|(T_K - T_n^t)r_t\right\|_{L^2(X,\rho)}^2 \leq 4\|f^*\|_{L^\infty(X,\rho)}^2\|K - K_0\|_{L^2(X^2,\rho\otimes\rho)}^2 + \epsilon. \tag{11}$$

Well by Lemma B.6 we have that $\|\theta_t - \theta_0\| \leq \frac{\sqrt{t}}{\sqrt{2}} \|f^*\|_{L^\infty(X,\rho)}$. Thus for $t \leq \frac{2R^2}{\|f^*\|_{L^\infty(X,\rho)}^2}$ we have that $\|\theta_t - \theta_0\| \leq R$. Well then by Lemma D.2 and the inequality (11) we have that

$$\|P_k(r_t - \exp(-T_K t)r_0)\|_{L^2(X,\rho)}^2$$

$$\leq \left[\frac{1 - \exp(-\sigma_k t)}{\sigma_k}\right]^2 \cdot \left[4\|f^*\|_{L^\infty(X,\rho)}^2 \|K - K_0\|_{L^2(X^2,\rho\otimes\rho)}^2 + \epsilon\right]$$

and

$$\|r_t - \exp(-T_K t)r_0\|_{L^2(X,\rho)}^2 \leq t^2 \cdot \left[4\|f^*\|_{L^\infty(X,\rho)}^2 \|K - K_0\|_{L^2(X^2,\rho\otimes\rho)}^2 + \epsilon\right].$$

The desired result then follows by setting $T = \frac{2R^2}{\|f^*\|_{L^\infty(X,\rho)}^2}$. $\qquad\square$

# E  Discussion of Assumption 3.6

We will discuss why it is reasonable to assume that $m = \tilde{\Omega}(\epsilon^{-2})$ suffices to ensure that $\|K_0 - K^\infty\|_{L^2(X\times X,\rho\otimes\rho)}^2 \leq \epsilon$ holds with high probability over the initialization. We note that for fixed $\theta_0$, $K_0$ and $K^\infty$ are bounded and thus by Hoeffding's inequality we have that with high probability

$$\|K_0 - K^\infty\|_{L^2(X\times X,\rho\otimes\rho)}^2$$

$$\leq \frac{1}{N}\sum_{i=1}^N |K_0(x_i, x_i') - K^\infty(x_i, x_i')|^2 + \tilde{O}\left(\frac{\|K_0 - K^\infty\|_{L^\infty(X\times X,\rho\times\rho)}^2}{\sqrt{N}}\right),$$

where $(x_1, x_1'), \ldots, (x_N, x_N')$ is an i.i.d. sample from $\rho \otimes \rho$. Furthermore we have by Lemma B.4 that $\|K_0 - K^\infty\|_{L^\infty(X\times X,\rho\times\rho)}^2 = \tilde{O}(1)$ with high probability over the initialization of $\theta_0$. Thus if we set $N = \tilde{\Omega}(\epsilon^{-2})$ we have that Assumption 3.6 holds provided that

$$\frac{1}{N}\sum_{i=1}^N |K_0(x_i, x_i') - K^\infty(x_i, x_i')|^2 = O(\epsilon)$$

with high probability over the simultaneous sampling of $\theta_0$ and $(x_1, x_1'), \ldots, (x_N, x_N')$.

It is been shown in many settings that the pointwise deviations satisfy

$$|K_0(x, x') - K^\infty(x, x')| = \tilde{O}(1/\sqrt{m})$$

with high probability over $\theta_0$. The earliest was Du et al. (2019b) who demonstrate that for a shallow ReLU network for fixed $x, x'$ we have with probability at least $1 - \delta$ over the initialization

$$|K_0(x, x') - K^\infty(x, x')| \leq O\left(\frac{\log(1/\delta)}{\sqrt{m}}\right).$$

Analyzing the portion of the Neural Tangent Kernel corresponding to the last hidden layer, Du et al. (2019a) get an analogous bound for deep fully-connected, ResNet, and convolutional networks with smooth activations. This is substantiated by the results of Huang & Yau (2020) for deep fully-connected networks with smooth activations. In their work they demonstrate that for a fixed training set $x_1, \ldots, x_n$

$$\max_{i,j} |K_0(x_i, x_j) - K^\infty(x_i, x_j)| = \tilde{O}(1/\sqrt{m})$$

with high probability over the initialization. In their result there are constants that depend on how well dispersed $x_1, \ldots, x_n$ are. Bowman & Montúfar (2022) demonstrated that for shallow fully-connected networks with smooth activations

$$\sup_{(x,x')\in X\times X} |K_0(x, x') - K^\infty(x, x')| = \tilde{O}(1/\sqrt{m})$$

with high probability over the initialization. For deep fully-connected ReLU networks Arora et al. (2019b) demonstrate that for fixed $x, x'$ if $m = \Omega(L^6 \log(L/\delta)/\epsilon^4)$ then with probability at least $1 - \delta$

$$|K_0(x, x') - K^\infty(x, x')| \leq (L+1)\epsilon.$$

In terms of the width $m$ this translates to $|K_0(x,x') - K^\infty(x,x')| = \tilde{O}(1/m^{1/4})$ with high probability. This was improved in a recent work by Buchanan et al. (2021) that demonstrated that if $\mathcal{M}$ is a Riemannian submanifold of the unit sphere then with high probability over the initialization

$$\sup_{x,x'\in\mathcal{M}\times\mathcal{M}} |K_0(x,x') - K^\infty(x,x')| = \tilde{O}(1/\sqrt{m}).$$

Furthermore as stated by Buchanan et al. (2021) their analysis should be amenable to other architectures.

Now note that $\max_{i\in[N]} |K_0(x_i,x_i') - K^\infty(x_i,x_i')| = O(\epsilon^{1/2})$ suffices to ensure that

$$\frac{1}{N}\sum_{i=1}^{N} |K_0(x_i,x_i') - K^\infty(x_i,x_i')|^2 = O(\epsilon).$$

Based on the previous discussion, we expect that with high probability

$$\max_{i\in[N]} |K_0(x_i,x_i') - K^\infty(x_i,x_i')| = \tilde{O}(1/\sqrt{m}).$$

Thus if $m = \tilde{\Omega}(1/\epsilon^2)$ then we would have that $\max_{i\in[N]} |K_0(x_i,x_i') - K^\infty(x_i,x_i')| = \tilde{O}(\epsilon)$ which is stronger than what we need. In fact $\max_{i\in[N]} |K_0(x_i,x_i') - K^\infty(x_i,x_i')| = \tilde{O}(1/m^{1/4})$ is sufficient. For these reasons, we view Assumption 3.6 as quite reasonable. Nevertheless, we are not aware of an out-of-the box result that simultaneously addresses all the cases we consider and thus we must add this as an external assumption. However, if desired one can bypass Assumption 3.6 by citing the aforementioned results to get statements for the cases in which they apply to.

## F Experimental Details

**Architecture and Parameterization** The code to produce Figure 1 is available at `https://github.com/bbowman223/deepspec` The NTK Gram matrix $(G_0)_{i,j} := K^{\theta_0}(x_i,x_j) = \langle\nabla_\theta f(x_i;\theta_0), \nabla_\theta f(x_j;\theta_0)\rangle$ was computed for two separate networks. The first network corresponds to LeNet-5 (LeCun et al., 1998) where the output is the logit corresponding to class 0. The second network is a feedforward network with one hidden layer with the Softplus activation $\omega(x) = \log(1 + \exp(x))$. For LeNet-5 we compute the NTK using PyTorch (Paszke et al., 2019) using the default PyTorch initialization and parameterization. For the shallow network we implement the network directly and use the Neural Tangent Kernel parameterization:

$$f(x;\theta) = \frac{1}{\sqrt{m}}\sum_{i=1}^{m} a_i\omega(\langle w_i, x\rangle + b_i) + b_0,$$

where there is an explicit $1/\sqrt{m}$ factor. All parameters for the shallow network are initialized as i.i.d. standard Gaussian random variables $N(0,1)$.

**Details of Computation** For each network we compute the NTK Gram matrix $G_0$ for 10 separate pairs of $(\theta_0, S)$ where $\theta_0$ is the parameter initialization and $S = (x_1,\ldots,x_n)$ is the data batch. Each line in the plots of Figure 1 corresponds to a different pair $(\theta_0, S)$. We simultaneously sample the parameter initialization $\theta_0$ and a random batch of 2000 training samples $x_1,\ldots,x_{2000}$. We load the batches using "DataLoader" in PyTorch with the "shuffle" parameter set to True. This means the batches will be sampled sequentially from a random permutation of the training data and thus are sampled without replacement. We then compute the NTK Gram matrix $(G_0)_{i,j} := K^{\theta_0}(x_i,x_j) = \langle\nabla_\theta f(x_i;\theta_0), \nabla_\theta f(x_j;\theta_0)\rangle$. Once we compute $G_0$ we compute its spectrum and plot the first 1000 eigenvalues. Note that the number of eigenvalues that we plot is half the batch size. We observe that if one plots all $n$ eigenvalues (the number of eigenvalues equals the number of samples) one gets a sharp drop in log scale magnitude starting near the bottom 5-10% of eigenvalues. We observed this to occur even as one varies $n$. We suspect this is due to numerical errors and thus we only plot the first half of the spectrum.

**Data** The dataset used for LeNet-5 is MNIST (LeCun et al., 1998) and the dataset for the shallow model is CIFAR-10 (Krizhevsky, 2009). MNIST is made available through the Creative Commons Attribution-Share Alike 3.0 license. CIFAR-10 does not specify a license. Neither of these datasets have personally identifiable information nor offensive content.

**Computational Resources and Runtime**   The experiments were run on a 2016 Macbook Pro with a 2.6 Ghz Quad-Core Intel Core i7 processor and 16GB of RAM. The experiment took less than an hour in wall-clock time.

**Software Licenses and Attribution**   Our experiments were implemented in Python with the aid of the following software libraries/tools: PyTorch (Paszke et al., 2019), NumPy (Harris et al., 2020), SciPy (Virtanen et al., 2020), Matplotlib (Hunter, 2007), Jupyter Notebook (Kluyver et al., 2016), IPython (Pérez & Granger, 2007), and autograd-hacks `https://github.com/cybertronai/autograd-hacks`. PyTorch, Numpy, and SciPy are available under the BSD license. Jupyter and IPython are available under the new/modified BSD license. Matplotlib uses only BSD compatible code and is available under the PSF license. The code for autograd-hacks belongs to the public domain as specified by the public-domain-equivalent-license "Unlicense" `https://unlicense.org/`.