# OpenReview forum: "Spectral Bias Outside the Training Set for Deep Networks in the Kernel Regime"
_NeurIPS.cc/2022/Conference — NeurIPS 2022 Accept_

### Official Review · Reviewer_9oZz · 2022-07-09

**Rating:** 6
**Confidence:** 4
**Soundness:** 3 good
**Presentation:** 3 good
**Contribution:** 3 good

**Summary:**

This article theoretically offers quantitative bounds between the trajectory of a finite-width network from the idealized kernel dynamics of infinite one, which implies that networks tend to learn the top eigenfunctions of the NTK at rates corresponding to their eigenvalues. Furthermore, the authors observe that if the width of network grows linearly with the number of samples, it will remain the bias of the kernel at beginning.

**Questions:**

1.	It seems that this article does not consider the bias term in the neural network. If the bias term is considered, do the corresponding results still hold?
2.	Many Relu-like activation functions do not satisfy Assumption 3.1. Can authors discuss more about such activation functions?
3.	If I understand correctly, as t goes to infinity, the RHS of the first inequality converges to a non-zero constant. Does mean this estimation is too loose. And, how can this inequality relates the behavior of the convergence outside the training set?
4.	In “NTK Eigenvector and Eigenfunction Convergence Rates”, a related reference is missed: https://arxiv.org/abs/2010.08153


**Limitations:**

1.	The main limitation of this paper is that although the theory is rich, there are too few experiments to tell the reader at a glance what it does. The interpretation of theoretical result is not clear enough, especially, for “outside the training set”.

**Strengths And Weaknesses:**

Strengths:
1.	The paper is rich in theory and offers quantitative bounds between the trajectory of a finite-width network from the idealized kernel dynamics of infinite one, which measure the L2 difference in function space and imply that networks tend to learn the top eigenfunctions over the entire input space. I'm sure the audience of NeurIPS will be interested in it.
2.	The theoretical proof is detailed, the order is reasonable and the structure is clear.

Weaknesses:
1.	Although the theory of this article is rich, there are very few experiments that could be found in it, which could verify the rationality of the hypothesis and whether the theoretical derivation is true or not. One suggestion is that: It will make the conclusions more convincing if corresponding graphs of experimental results could be attached after important conclusions. Especially, authors can use a 1d experiments to show the estimation in theorem 1 is meaningful.

---

> ### Author Response · Authors · 2022-08-02
> **Initial response to reviewer 9oZz (part 1)**
>
> First, thank you for engaging with our work and bringing up some important questions.  Your feedback about adding experimental illustrations is helpful.  While we did choose to pursue this theoretical direction in part based on experiments that were shared with us by some of our collaborators, it is true that adding some experiments could aid the reader to interpret the result better.  We are pursuing adding experiments to an updated version of the manuscript.  We will add a comment if we are able to upload the updated version before the deadline.
>
> As for your question "If the bias term is considered, do the corresponding results still hold?", the answer is we believe so with high confidence.  The reason we do not consider the bias terms is because we make use of the Hessian bound in [1] which assumes the network does not have bias terms.  However, we believe with high confidence that their proof goes through for the case where the network has biases.  Another point that adds confidence is that in [4] for the shallow case the authors are able to consider biases in the network.
>
> Your second question "Many Relu-like activation functions do not satisfy Assumption 3.1. Can authors discuss more about such activation functions?" is a relevant one.  Assumption 3.1 is also necessary in order for us to use the Hessian bound in [1].  To respond to this question, we refer the reviewer to our discussion about ReLU in the second paragraph in our response to reviewer 32uw: Indeed the requirement that the activation function be twice differentiable prevents the theorem from applying to ReLU.  The result from [1] bounding the Hessian norm for the network that we utilize requires this assumption.  Without modifying the theorem the closest we could get to ReLU would be to use a softmax approximation to ReLU, say $\frac{1}{\alpha} \log(1 + \alpha x)$ for some $\alpha > 0$.  As $\alpha \rightarrow \infty$, there are constants in the result from [1] that will grow that will increase the requirement on the width.  Unfortunately we are not aware of any out-of-the-box results that would permit our theorem to extend to ReLU immediately.  However if one were to extend the results in [1] to the ReLU activation (which would be worthy of a paper on its own) the rest of the proof should go through.
>
> We now address the third question "If I understand correctly, as t goes to infinity, the RHS of the first inequality converges to a non-zero constant. Does mean this estimation is too loose.".  In Theorem 3.5 it is true that the first term will be nonvanishing for general kernels $K$.  However, in the standard case $K = K^\infty$ we see in Corollary 3.7 that the bound converges to $\epsilon / \sigma_k^2$ which can be made to vanish by sending $\epsilon \rightarrow 0$.  Setting $K = K^\infty$ is the most natural choice since $K_0$ converges to $K^\infty$ as the width of the network grows.  The reason we allow for an arbitrary kernel $K$ instead of only considering $K^\infty$ is because occasionally there are other kernels that people consider other than $K^\infty$ such as the conjugate kernel (which corresponds to training only the outer layer).  We introduce Theorem 3.5 in full generality to allow for these comparisons.
>
> As for your question "how can this inequality relate the behavior of the convergence outside the training set", the reason our statement is meaningful outside the training set is because we track the $L^2$ error with respect to the true input distribution $\rho$ which covers all inputs, not just those in the training set.  In contrast to the empirical error which is measured by $L^2$ with respect to the empirical measure, the generalization error is measured with respect to $\rho$.  Thus when we say "off the training set" we mean we are characterizing the error in terms of the true input distribution which covers the entire input domain (not just the training set).  We note that in Corollary 3.10 we obtain a bound on the test/generalization error.  This would be impossible if we were not using the true input distribution $\rho$ and were restricted to the training set.  While previous works such as [2], [3] track the error on the training set and track the empirical risk, our work is able to describe the behavior over the whole input domain and consequently the test/generalization error.  There was one notable work [4] which operated beyond the training set, however they require that the network is shallow and underparameterized.  We were able to remove these unrealistic conditions to allow for deep networks with more moderate sample complexities.  The fact that we measure the $L^2$ distance with respect to the true input distribution $\rho$ which characterizes the error outside the training set is one of the most essential contributions of our work.  We hope that this important fact is taken into account in your final review.

---

> ### Author Response · Authors · 2022-08-02
> **Initial response to reviewer 9oZz (part 2)**
>
> As for your comment about the reference "On the exact computation of linear frequency principle dynamics and its generalization", we think this is an interesting work and are glad you pointed it out.  It is quite nice that they are able to track the dynamics of the infinite width two layer model in frequency domain.  This gives a nice explicit description of "Spectral Bias" in the sense of Fourier Frequencies.  **We have updated our manuscript to cite this relevant work.**
>
> We again thank you for your helpful comments.  Your review has helped us improve our manuscript.  We have made careful efforts to address all your concerns and comments.  We hope that the clarification of our bound as well as the essential detail of "on the training set" versus "off the training set" will not be overlooked and will be taken into account in your final score/evaluation.  Please let us know if you have further questions; thank you for your valuable time.
>
> [1] Chaoyue Liu, Libin Zhu, Mikhail Belkin. "On the linearity of large non-linear models: when and why the tangent kernel is constant" https://arxiv.org/abs/2010.01092
>
> [2] Arora et al. "Fine-grained analysis of optimization and generalization for overparameterized two-layer neural networks." https://arxiv.org/pdf/1901.08584.pdf
>
> [3] Cao et al. "Towards understanding the spectral bias of deep learning" https://arxiv.org/abs/1912.01198
>
> [4] Benjamin Bowman and Guido Montufar. "Implicit Bias of MSE Gradient Optimization in Underparameterized Neural Networks" https://arxiv.org/abs/2201.04738

---

### Official Review · Reviewer_XnSP · 2022-07-11

**Rating:** 8
**Confidence:** 4
**Soundness:** 4 excellent
**Presentation:** 4 excellent
**Contribution:** 3 good

**Summary:**

The paper bounds the distance between the dynamics of wide DNNs and the dynamics of kernel gradient descent on the population with the limiting NTK. This paper improves over previous bounds, by considering the projected error along the top k eigenfunctions of the integral operator $T_{K^\infty}$ for any $k$.

They show that the approximation is better along the top eigenvalues. This allows them to show that with early stopping, one can guarantee learning along the top eigenvalues of DNNs. In comparison to previous bounds, the results in this paper only require a width that grows linearly with the number of datapoints, at least with a specific early stopping.

**Questions:**

I do not have any questions or suggestions regarding the aforementioned weakness (except to try and do the proofs for gradient descent but I realize that it would require a lot of work).

**Limitations:**

As already mentioned I think the paper would be greatly improved with a discussion of gradient descent.

**Strengths And Weaknesses:**

The paper is well written and gives a good overview of the previous bounds and how their bound improves on previous ones. The paper nicely connects two lines of research: the NTK regime approximations for finite widths and the spectral bias of DNNs.

The results are new to my knowledge and describe an interesting phenomenon. I did not check the proofs in detail but I did not see any technical issues.

The only weakness that concerns me is the fact that the results only apply to gradient flow. The discussion of early stopping is much more interesting in the context of gradient descent since it then relates to computational complexity. There exist many proofs of convergence to NTK-dynamics for gradient descent, so I think the proofs should extend to this setting easily. I do not understand why the authors did not try to extend their proof to gradient descent.

---

> ### Author Response · Authors · 2022-08-02
> **Initial Response to Reviewer XnSP**
>
> First, thank you for giving our paper a careful read and conscientious review.  You bring up an interesting point about gradient descent versus gradient flow.  Typically gradient flow is seen as a fair approximation to gradient descent whenever the step size is small, however gradient descent proofs do introduce additional technicalities.  While we believe one could likely prove an analogous statement for gradient descent, the proof would become far more complicated.  A big benefit of using gradient flow is one has the nice "Damped Deviations" equation (originally introduced in [1]) that appears in Lemma D.1.  This "Damped Deviations" point-of-view of analyzing the dynamics from an ODE perspective makes the optimization dynamics much more intuitive.  However, when analyzing the empirical risk the authors in [2] have in equation (27) an expansion of the residual (in their notation $u(k) - y$) that can be viewed as a discrete-time version of the "Damped Deviations" equation restricted to the training set.  This equation is more complicated than the continuous time version, however it gives confidence that there is some precedence for discrete time analysis of convergence from this angle.  While gradient descent analysis was outside the scope of our present work, we do agree it is ultimately an interesting line to pursue.
>
> Thank you for your helpful feedback and careful review.  Please let us know if you have any further questions.
>
> [1] Benjamin Bowman and Guido Montufar. "Implicit Bias of MSE Gradient Optimization in Underparameterized Neural Networks" https://arxiv.org/abs/2201.04738
>
> [2] Arora et al. "Fine-grained analysis of optimization and generalization for overparameterized two-layer neural networks." https://arxiv.org/pdf/1901.08584.pdf

---

### Official Review · Reviewer_psNV · 2022-07-11

**Rating:** 6
**Confidence:** 3
**Soundness:** 2 fair
**Presentation:** 3 good
**Contribution:** 2 fair

**Summary:**

This paper provides a new theoretical analysis of the bounds of function space difference between a finite-width network in a finite sample size regime and the corresponding kernel dynamics in finite width and finite data regime.
The results are presented based on fully connected, convolutional and residual neural network architectures.
By investigating the spectrum of the NTK and its connections to the Fisher Information Matrix, this work provided some insights into the spectral bias of neural network learning dynamics.

**Questions:**

- Besides the presented theoretical results, have the authors tried on any synthetic or MNIST-scale datasets to see if the results are consistent with the empirical experimental results?

**Limitations:**

N/A.

**Strengths And Weaknesses:**


#### Strengths
- This work continues to explore the spectral bias of the neural networks from the perspective of the neural tangent kernel theory. This work made some novel theoretical analyses on measuring the distance in function space and the scaling properties of the learning with respect to the stopping time.
- This paper is generally clearly written. Alongside the main theoretical results, it provides some description and interpretation of the main theorems.

#### Weaknesses
- My main concerns are about the lack of more discussions about the connections with existing literature ([1,2,3,4,5]) on analyzing the spectral structure of neural networks and corresponding kernel formulations, for example, existing works [1] provided theoretical results on the spectral alignment of neural network and kernel methods in learning dynamics, [2] addressed the low-rank spectral structure of both neural tangent kernel and the Fisher information matrix. Not limited to the missing references below, a more detailed discussion on the theoretical and empirical connections to the relevant literature is needed.

[1] Atanasov, Alexander, Blake Bordelon, and Cengiz Pehlevan. "Neural networks as kernel learners: The silent alignment effect." arXiv preprint arXiv:2111.00034 (2021).

[2] Zhang, R., Zhai, S., Littwin, E., & Susskind, J. (2022). Learning Representation from Neural Fisher Kernel with Low-rank Approximation. arXiv preprint arXiv:2202.01944.

[3] Karakida, Ryo, Shotaro Akaho, and Shun-ichi Amari. "Pathological spectra of the fisher information metric and its variants in deep neural networks." arXiv preprint arXiv:1910.05992 (2019).

[4] Pennington, Jeffrey, and Pratik Worah. "The spectrum of the fisher information matrix of a single-hidden-layer neural network." Advances in neural information processing systems 31 (2018).

[5] Hazan, Tamir, and Tommi Jaakkola. "Steps toward deep kernel methods from infinite neural networks." arXiv preprint arXiv:1508.05133 (2015).

---

> ### Author Response · Authors · 2022-08-02
> **Initial Response to Reviewer psNV (part 1)**
>
> Thank you for taking the time to review our paper.  We appreciate your comment on the novel contribution of our bounds, as well as your comment that the paper is clearly written.
>
> We must admit that we have some confusion about your review.  The only weakness that you list is that we are lacking a discussion of the relevant literature, so much so that you say a reject rating is warranted.  We find this characterization of our work inaccurate; we emphasize **that we cite over 50 works, mention 3 different research areas in the "Related Work Section", as well as have a separate section "Technical Comparison to Prior Work" in the main text.**  We take pride in giving proper credit to prior work and made extensive efforts to address the literature in our manuscript.
>
> Please keep in mind that it is impossible for us to cite every work that has some relation to our work (citing over 50 works is already much larger than a normal manuscript).  None of the papers you list overlaps significantly with the problems we address in our current manuscript.  We address each of the works you mention below.
>
> [1] describes the adaptations of the NTK for homogenous networks when the scale of the initialization is small and the data are whitened.  Furthermore they demonstrate that the learned neural network function at the end of training is well described by the NTK at the end of training (it behaves like the function given by the representer theorem in kernel ridge regression). This is an excellent work, however it addresses a different problem.  While we characterize the extent to which the early stage of training is described by the kernel dynamics with respect to the fixed kernel $K^\infty$, this work focuses on how the kernel adapts and how the final kernel at the end of training characterizes the final neural network solution.  Although it does not address an overlapping problem with our current manuscript, **we have added a citation of this work when we discuss kernel adaptations in the conclusion in the "Limitations" section.**
>
> [2] introduces a new kernel, the Neural Fisher Kernel (NFK), as a kernel that can address both unsupervised and supervised settings.  They exploit the low-rank property of the NFK to efficiently get low-dimensional feature representations from this kernel using automatic differentiation tools.  This paper does not attempt to address characterizing the optimization trajectory.  The overlap with our work is the fact that they also exploit the low-rank structure of the NTK/FIM.  However, as we mention in the manuscript the low-rank structure of the NTK/FIM is a well known fact.  Outlier eigenvalues of the Hessian (which is often approximated by the FIM) were observed as early as 1991 in "Eigenvalues of covariance matrices: Application to neural-network learning" by Le Cun et al.  There are more works that observe this fact over the following three decades than we could ever hope to address.  Our contribution in this manuscript was not to introduce this well-known fact, rather to provide novel quantitative bounds on the optimization trajectory.
>
> [3] theoretically studies the spectrum of the FIM for deep fully-connected networks.  They asymptotically characterize the mean, second-moment, as well as the top and second eigenvalue.  Consequently they are able to demonstrate outlier spectra, which they also verify empirically.  We note that our work does not attempt to characterize the spectrum of the NTK/FIM.  While the spectrum of the NTK/FIM is highly relevant for our results, we focus on quantitative bounds for the optimization trajectory of the network in function space.  Our plot of the NTK spectrum in Figure 1 is illustrative and is not presented or emphasized as a fundamental contribution, since the outlier spectra phenomenon has been observed empirically in many works as we mention in the manuscript.  Nevertheless, **we have decided to add a reference to this and a few other theoretical works providing further context to the list of works referenced in the related works "Spectrum of the NTK/Hessian and Generalization" section.**
>
> [4] uses tools from Random Matrix Theory to characterize the limiting spectral density of the Fisher Information Matrix of a single hidden layer neural network.  This is an interesting work but it is addressing a much different problem.  Again we reiterate our prior comments: our work does not address characterizing the spectrum of the NTK or FIM.  While the spectrum of the NTK/FIM is relevant for our work, characterizing the spectrum was not something that we address in this manuscript.  Rather, we focus on characterizing the optimization trajectory of the neural network in function space.  However, we did find some readers will be interested in works that theoretically analyze the spectrum so **we have added a citation for this work (among several others that are relevant).**

---

> > ### Comment · Reviewer_psNV · 2022-08-09
> > **Thanks for the response**
> >
> > I would like to thank the authors for the detailed response, which is quite helpful. My concerns are addressed, and I have updated my review to reflect the support.

---

> ### Author Response · Authors · 2022-08-02
> **Initial Response to Reviewer psNV (part 2)**
>
> [5] introduces kernels that describe the covariances of the Gaussian processes associated with infinitely wide neural networks.  They also offer a generalization bound in their Theorem 2 to explain how with proper regularization such networks don't overfit.  While indeed the connection of infinitely wide neural networks to Gaussian processes is a rich area that goes all the way back to Neal 1995 "Bayesian learning for neural networks", the connection between infinitely wide neural networks and Gaussian processes is not an angle we pursue.  Again this goes back two decades and thus it would be impossible to cite all the works in this area, some examples include "Deep Neural Networks As Gaussian Processes" by Lee et al., "Kernel Methods for Deep Learning" by Cho and Saul, "Gaussian Process Behaviour in Wide Deep Neural Networks" Matthews et al., "Bayesian Deep Convolutional Networks with Many Channels are Gaussian Processes" Novak et al., etc.  We do not think the Gaussian process angle connects with our work enough to make it useful to the reader to enumerate a list of such works.
>
> We think it is highly evident that works [1], [2], [3], [4], [5] do not address significantly overlapping problems that warrant a reject review on the basis of prior work.  **We conducted an extensive and professional literature review: we cited over 50 works, mention 3 different research areas in the "Related Work Section", as well as have a separate section "Technical Comparison to Prior Work" in the main text.**  We hope that our explanation of the works above in relation to our work will motivate you to change your recommendation.  In the case of further disagreement, we will defer to the area chair to decide if a reject rating can be justified on this basis.
>
> As for your question "Besides the presented theoretical results, have the authors tried on any synthetic or MNIST-scale datasets to see if the results are consistent with the empirical experimental results?", we are pursuing adding experiments to an updated version of the manuscript.  We will add a comment if we are able to upload the updated version before the deadline.

---

### Official Review · Reviewer_32uw · 2022-07-12

**Rating:** 7
**Confidence:** 2
**Soundness:** 3 good
**Presentation:** 3 good
**Contribution:** 3 good

**Summary:**

This paper presents a theoretical analysis
- bounding the difference between a kernel dynamics of finite width network trained for time T with a finite set of samples and the ideal infinite width NTK kernel.
- that concludes a finite width network is biased toward learning the top eigenfunction of the infinite width-based kernel

**Questions:**

- The theorem associates the minimum width of the network and the number of samples to the number of iterations trained. I'm confused by this connection - scaling the size of the network is possible but the size of the data is, in practice, fixed. Does this mean that for fixed-size data, the results are applicable only up to a certain training step?


**Limitations:**

- The paper targets a simple question of spectral bias of networks using NTK kernels and provides a characterization that is applicable to deep networks.
- The dependency on stopping time limits the consequences of the analysis to when the network is initialized (with few training steps)

**Strengths And Weaknesses:**

- The paper positions itself well amongst related works and is presented such that even an outsider to the topic is able to understand the problem, analysis, and results.
- The paper is constrained to bounding the difference in the kernels and it is not clear how the network architecture and activation function fits into the analysis. A discussion on how the model design affects the results could make clear the limitation as well as applicability to future model designs.
- The analysis considered requires twice differentiable activation function which does not include ReLU. I think this is an admissible assumption for theoretical analysis though it makes analysis disconnected from practice.
Minor comment
- the notation for non-linearity and eigenvalue seems to be overloaded and might create confusion.

---

> ### Author Response · Authors · 2022-08-02
> **Initial Response to Reviewer 32uw (part 1)**
>
> First, thank you for giving our paper an honest read and a thoughtful review.  Your first point about model design is quite an important one.  Our result characterizes how the spectrum of the NTK will affect the learning dynamics over the entire input space.  A natural question to ask then is how does the parameter initialization and model architecture influence the NTK spectrum.  Some activation functions, data sets, and architectures will have a more skewed spectrum than others.  For example, in classification problems the NTK tends to have a number of outliers on the order of the number of classes.  There is a rich line of work studying the NTK spectrum that can help inform how to apply our results.  As a concrete example, if one knows the number of outlier eigenvalues of the NTK, one can determine how many components of the NTK will be learned quickly at moderate sample and width complexities, as our bound is sharpest along the top components.  **Based on your feedback, we have now added a new list of works to the related work section that discuss the spectrum of the neural tangent kernel in the segment titled "Spectrum of the NTK/Hessian and Generalization".**
>
> Indeed the requirement that the activation function be twice differentiable prevents the theorem from applying to ReLU.  The result from [1] bounding the Hessian norm for the network that we utilize requires this assumption.  Without modifying the theorem, the closest we could get to ReLU would be to use a softmax approximation to ReLU, say $\frac{1}{\alpha} \log(1 + \alpha x)$ for some $\alpha > 0$.  As $\alpha \rightarrow \infty$, there are constants in the result from [1] that will grow and will increase the requirement on the width.  Unfortunately we are not aware of any out-of-the-box results that would permit our theorem to extend to ReLU immediately.  However if one were to extend the results in [1] to the ReLU activation (which would be worthy of a paper on its own) the rest of the proof should go through.
>
> Your comment on the overloaded notation for the non-linearity and eigenvalues was helpful.  **We will update the notation for the nonlinearity to eliminate this confusion.**
>
> As for your question (fixed-size data vs training step), it is true that if the number of samples are fixed the result is only meaningful up to a certain stopping time (once you fix the error tolerance $\epsilon$).  Prior work [2] required the number of samples to be larger than the number of parameters, which does not hold in practice as most modern networks are overparameterized.  Our contribution in this regard was to relax the sample complexity to allow for overparameterization.  While it is true that in practice one has the ability to scale the network size within compute budget limits, the typical NTK regime where the network width is polynomially wide relative to the number of samples is still unrealistic relative to modern practice.  Our contribution in that regard was to form a statement that could be meaningful even when the network width is on the same order as the number of samples.  However, if one is willing to operate in the regime where the network is polynomially wide relative to the number of samples we believe that one could possibly remove the stopping time and prove an alternative version of our theorem.  The reason for this is that we use the stopping time to bound the parameter deviations from the initialization, however if you are in the regime where the loss converges to zero you get bounded parameter deviations as a result of convergence.  Thus if you are able to assume convergence you do not need to employ a stopping time in order to bound the parameter deviations.  Since our focus was on the regime where the network width is more moderate, we ultimately did not pursue this direction.  However it is an interesting direction to pursue.
>
> In regard to your comment that the dependency on the stopping time limits the consequences of the analysis to the early stages of training, we do admit that this is a limitation.  It is for this reason that we made sure to mention this point in the "Limitations" section.  Employing a stopping time was the trade-off we needed to make in order for our result to hold for moderate widths and to avoid the standard NTK regime of polynomially wide networks.  However as we mention in the previous paragraph, there are ways to avoid a stopping time if one is willing to operate in the polynomially wide regime.  Since prior work has addressed the polynomially wide regime well, we ultimately chose to pursue the other direction.

---

> ### Author Response · Authors · 2022-08-02
> **Initial Response to Reviewer 32uw (part 2)**
>
> Thank you again for giving the paper a careful read and a thoughtful review.  We have made careful efforts to address all of your comments and concerns.  Your comments have helped us improve the manuscript.  We hope that our responses clarify the work and address any concerns you have.  Please let us know if you have any further questions.
>
> References:
>
> [1] Chaoyue Liu, Libin Zhu, Mikhail Belkin. "On the linearity of large non-linear models: when and why the tangent kernel is constant" https://arxiv.org/abs/2010.01092
>
> [2] Benjamin Bowman and Guido Montufar. "Implicit Bias of MSE Gradient Optimization in Underparameterized Neural Networks" https://arxiv.org/abs/2201.04738

---

> > ### Comment · Reviewer_32uw · 2022-08-09
> > **Comment on rebuttal.**
> >
> > Thanks for the response and clarifications. Good work!

---

### Meta-Review · Area_Chair_1dnU · 2022-08-26

**Recommendation:** Accept
**Confidence:** Certain

**Metareview:**

This paper focuses on theoretically bounding the difference between a finite-width network in a finite sample size regime and the corresponding kernel dynamics in finite width and finite data regime for various neural network architectures. Using the spectrum of the NTK they provide some insights into the spectral bias of neural nets. All reviewers were positive and recommend acceptance. I concur with this decision.


**Award:**

No

---

### Decision · Program_Chairs · 2022-09-14

Accept